# Post-COVID-19 Pandemic Era and Sustainable Healthcare: Organization and Delivery of Health Economics Research (Principles and Clinical Practice)

**Mohammad Heydari [1],* and Kin Keung Lai [2]**

1 School of Economics and Management, Tongji University, Shanghai 200092, China
2 International Business School, Shaanxi Normal University, Xi'an 710062, China; mskklai@outlook.com
* Correspondence: mohammadheydari1992@yahoo.com

**Abstract:** Health services research aims to improve population health by studying the organization, delivery, and financing of healthcare services. While the field has made progress in defining its boundaries and core research topics, our literature survey revealed a lack of attention given to the management, cost, and policy aspects of healthcare systems (SCs). Moreover, the readiness of the system to handle supply policy and device deficiencies, especially during the pandemic, was rarely mentioned. Unplanned urban growth, characterized by reduced open spaces, deteriorating infrastructure, and changes in biological morphology, has led to an uneven distribution of urban amenities, facilities, and healthcare services. This research proposes two reliable models for site selection in a major hospital in Hong Kong, considering uncertainty levels, infeasibility tolerance, and reliability. We examine two categories of uncertainty—symmetric and bounded—and provide a solution with a nominal objective function value of 121.37. By considering 23 uncertain parameters with specific tolerance levels, we extend the processing time of uncertain tasks to ensure feasibility. However, the objective function value decreases due to violations of intermediate due-dates and decreased overall production. A comparative analysis is presented to evaluate the solution and address scheduling challenges under uncertainty using a specified probability distribution function. The study concludes by introducing justice and health problems, outlining four typical strategies, and emphasizing the importance of the effective management of systems, components, and procedures for the production, distribution, and administration of medications and healthcare supplies. This research contributes to fairness in health systems and population health at local, national, and global levels, addressing health inequity and promoting public dialogues on the subject.

**Keywords:** post-COVID-19 pandemic era; health services research (health serv. res.); health economics; site selection; health and social policy evaluation; health disparities/inequalities and ethics; mixed-integer linear programming (MLIP); integer linear programming (ILP)

**MSC:** 90B50

## 1. Introduction

Researchers generally believe that the key to effectively ensuring the supply of emergency supplies lies in a rational Location Allocation Problem (LAP) and the scientific planning of the Vehicle Routing Problem (VRP). Moreover, there is an interdependent and interactional connection among LAP and VRP. Thus, it is necessary to design and optimize them as a whole system, that is, to study Location Routing Problems (LRPs) in the emergency logistics system. LAP issues mainly refer to the decision-maker recognizing the amount and location of facilities in a specific geographical area based on the geographical distribution of customers and goods.

Hospital delivery throughout the world has been fraught with expensive, lower-efficiency, and poorer-quality patient care services, and thus Hong Kong is no exception

in this regard. However, the qualities of the services provided for the patients have been far from exemplary. As an example, it is common that waiting times for specific routine surgeries at public hospitals last 18 months. Another exacerbated issue would be population aging; that is, the number of individuals aged $\geq$ 65 years of age is rapidly approaching 15% of the general population in Hong Kong, and this figure has been increasing annually by approximately 1%.

The hospital under present study is one of the major hospitals (MHs) in Hong Kong under Hospital Authority (HA) management. It provides residents of Hong Kong with a wide range of services. The MH offers both inpatient and specialized outpatient medical services. However, the current constraints of Queen Elizabeth Hospital (QEH) are summarized below:

➢ Hospital with a high volume of activities;
➢ Reached the limits of capacity;
➢ Heavily utilized and aged infrastructure.

Moreover, in line with the HA's speech at the Hospital Authority Convention 2013, the demand for hospital beds in Hong Kong will be about 2600 by 2021 and about 6600 by 2031. Therefore, the long-term plan of the government is to expand healthcare capacity. The MH plans to open a new special hospital ward (SHW) in Hong Kong. The most important issue is determining the right site for this new SHW.

The present study will represent a robust site selection model for the MH. The location of the hospital is critical in minimizing economic hardship for both the MH and the HA and maximizing the success of a Health Maintenance Organization (HMO) network, since the location of the hospital will definitely have a direct effect on HMO utilization [1]. However, in site selection, the fixed and variable costs and the government budget to operate the hospital are uncertain, unexpectedly increasing the complexity of decision-making.

To address the issue of parameter uncertainty, novel robust optimization protocol approaches have been developed. The robust optimization protocol would handle the parameter uncertainty, supposing that unknown factors belonged to a set of bounded convex uncertainties. Therefore, it is possible to minimize the negative impacts on the objective values and guarantee the solutions [2]. However, a basic concept behind the robust optimization protocol would be investigating the worst scenario with any particular distribution assumption. Notably, [3] has pioneered the works on the robust optimization protocol, which requires each uncertain parameter to reach its arrant case value to over-conserve the functional execution.

Moreover, [4–6] established several improvements with the ellipsoidal uncertainty set for adjusting conservatism levels and provided tractable mathematical reformulations. In addition, [7,8] investigated uncertainties via describing a polyhedron for all parameters. Consequently, they introduced the concept of "*budget-of-uncertainty*" for controlling conservation. Furthermore, [2] demonstrated that one range for all parameters would cause overly conservative outcomes and illustrated uncertain parameters with numerous ranges. Other works followed a framework based on a scenario wherein uncertainty was modelled using scenarios, which subsequently resulted in the stochastic programming formulas [9].

Nonetheless, obtaining precise probability data to distribute would be difficult, which varies timings. Another study conducted by [10] and [6] developed a powerful deterministic alternative regarding the amount of data uncertainties, tolerance of infeasibility, and levels of reliability, as the probabilistic measurement was applied. This protocol enjoys benefits such as linearity and applicability. It also could have a deterministically solved and easily controlled conservatism level. Therefore, the present robust optimization protocol model has been designed considering the work in the field described above and specifically applied to the site selection model for the MH.

The COVID-19 pandemic has not only posed significant challenges to healthcare systems worldwide but has also exposed critical gaps and shortcomings in the organization and delivery of healthcare services. As we navigate through the post-pandemic era, it becomes increasingly crucial to explore sustainable approaches in healthcare that prioritize

efficiency, accessibility, and resilience. In this context, health economics research plays a pivotal role in understanding the principles and practices necessary for building robust healthcare systems that can withstand future crises.

The aim of this study is to shed light on the gaps and importance of research focused on the organization and delivery of healthcare in the post-COVID-19 era, specifically from a health economics perspective. By examining the principles and clinical practices that underpin sustainable healthcare systems, we can identify strategies to enhance resource allocation, improve cost-effectiveness, and optimize health outcomes.

One of the key gaps that need to be addressed is the understanding of the long-term economic implications of the pandemic on healthcare systems. The unprecedented burden on healthcare resources, disruption of services, and shifting priorities have led to substantial economic consequences. Exploring the economic impact of the pandemic is essential for formulating evidence-based policies and allocating resources efficiently.

Moreover, the pandemic has highlighted the need for resilient healthcare systems that can adapt to sudden disruptions and maintain essential services. This calls for a deeper examination of health system financing, insurance mechanisms, and the role of public–private partnerships in ensuring sustainable healthcare delivery. By analyzing these aspects, we can develop frameworks and models that promote effective resource allocation and equitable access to healthcare services.

Furthermore, the pandemic has underscored the importance of preventive and primary healthcare, as well as the integration of technology and telemedicine in healthcare delivery. Understanding the economic implications and potential cost savings associated with these strategies is crucial for designing sustainable healthcare systems that prioritize preventive measures, early intervention, and an efficient use of resources.

In conclusion, as we move forward into the post-COVID-19 era, it is imperative to bridge the gaps in health economics research related to the organization and delivery of healthcare. By examining the principles and clinical practices that drive sustainable healthcare systems, we can identify innovative strategies, optimize resource allocation, and enhance the overall efficiency and effectiveness of healthcare delivery. This research aims to contribute to the body of knowledge necessary for building resilient and sustainable healthcare systems in a rapidly changing world.

## 2. A Review of the Literature

We strongly believe that incorporating a supply chain management (SCM) perspective into health service research offers investigators a valuable opportunity to access a wide range of theoretical frameworks that encompass interorganizational relationships, micro- and macroeconomics, regulation, intermediation, and even sociological aspects. Therefore, we express our gratitude for the article published in this issue by esteemed scholars in the field, which delves into a crucial and often underestimated component of health services: the SCs of medical devices. While the COVID-19 pandemic has heightened interest in the SCs, particularly in relation to the device sector and its management, exploration of this area predates the emergence of the virus. This commentary is founded on our comprehensive review of previous studies that shed light on the understanding of variation in medical device prices. It is also influenced by our own research experiences, which bridge the gap between SCs and health service research, as well as our firm conviction that adopting an SCs perspective can enhance the provision of health services in the post-COVID era [11].

### 2.1. Cardiac Care and Its Supply Chain as the Medical Devices

Health services research (Health Serv. Res.), an expanding field, has tried to specify its boundaries as a multidisciplinary discipline involving population wellbeing, access, quality, and cost [12]. While there is awareness of the significance of the impact of health technologies and health technologies assessment (HTA), in our examination of the Health Serv. Res. Literature, we found that SCs (supply chains) containing their cost, policy, management, and influence on outcomes, receive relatively little attention. Importantly,

our assessment also showed that the system's readiness to handle supply policy and supply device deficiencies like those encountered during COVID-19 was hardly ever mentioned.

We believe that infusing an SCM (supply chain management) concentration into Health Serv. Res. provides investigators with a nexus of theoretical frameworks integrating inter-organizational connections, micro- and macroeconomics, regulation, intermediation, and even a sociological component. Grennan et al. (2022) examine a vital but frequently disregarded aspect of healthcare services: medical devices SCs [13]. Even while the pandemic has increased interest in the SC, particularly in the device portion and its administration, research into this area began before the virus made an appearance [14]. Grennan et al. (2022) contribute to an understanding of price variation for medical devices, with our own research experiences bridging the gap between social science and Health Serv. Res. and our conviction that an SC perspective can enhance health services in the post-pandemic environment [13].

Effectively managing expensive medical goods, like implanted devices, has tremendous value for the health system because they are the second-largest expense category (after labor) [15]. At the same time, previous studies identified the variation in medical device prices [16] and their substantial variation [17]. Grennan et al. (2022) explore cardiac unit analysis, focusing on costs and management, posing it as especially essential in the quest to untangle the causes for large levels of cost variance across some cardiac products and relatively little variation in others [13,18].

The rise of the worldwide interventional cardiology market, which was worth USD 14 billion in 2019 and is anticipated to reach USD 16.2 billion in 2027, illustrates the significance of a concentration on cardiology device costs and cost variance [19]. We found significant price variations for stents between 2006 and 2014, with one set of categories being taken into account by Grennan et al. (2022). Moreover, bare-metal stent costs fell from roughly USD 1000 to just over USD 600, and drug-eluting stent prices fell from approximately USD 2300 to USD 1400 [13,20]. Stents' price variance is shown by Grennan et al. (2022) to be much lower than that of several other cardiac devices [13].

Gannon et al. (2019) contribute to understanding the underlying elements, such as variation [21]. They highlight the role of management in reducing costs in addition to a number of other factors, such as physician integration, standardization (as it is connected to a smaller supplier base), efficient value analysis processes that bring clinicians together to agree on products and commit to a brand, and so forth. Their realization that purchasing more of the same product from the same vendor at the same hospital matters for savings attests to appropriate supplier base reduction, a crucial concept in SCM. This highlights the significance of including general management practices and practices tied to strategic management progressive practices via SC scholars—practices that can lead to a more successful medical device SC—in research design [14]. We go into further detail about a few of these approaches, which, if applied to study design, can aid health services scholars in providing clarity to this urgently needed field of research—not just for cardiology but also for other high-supply-cost growth sectors like spine and orthopedics.

Moreover, [13]'s contention that better management practices are but one of several mechanisms to "*chip away at the large potential savings in hospital purchasing*", and that investments in management practices may be appropriate to achieve future savings, opens up the door for future health services and management studies into the mix of value-added management [22] practices and provides a reason for the further consideration of SCM investment via hospital leadership.

Their argument that improved management practices represent just one of several approaches to gradually reduce substantial potential savings in hospital purchasing, along with the suggestion that investments in management practices may be necessary to achieve future cost reductions, paves the way for future research in health services and management. This research can explore the combination of value-added management practices and provides a compelling rationale for hospital leadership to seriously consider investing in

supply chain management. In the subsequent Section 2.2, our focus shifts to examining various factors that impact the healthcare system's supply chain.

### 2.2. Supply Chain in Healthcare and Influencing Factors

#### 2.2.1. Innovation in Medical Devices

Burne's analysis (2012) shows that in several areas of cardiology, innovation has slowed down during the past ten years as hospitals and payors have become more cost-conscious [23]. This highlights the huge price decreases that have occurred in a number of cardiac implant types in recent years. While drug-eluting stents have replaced metal-bare stents as the industry standard, the cost of biodegradable stents has dramatically increased. The contrasts among "*commoditized items*" (those with few differences and various suppliers) are discussed with the right product and fewer alternatives that clarify pricing disparities. Future studies that examine pricing and the rates at which practitioners embrace new technology are urgently required throughout the spectrum of medical device categories, including orthopedics and spine. It is also vital to realize that breakthroughs, such as those allowing for minimally invasive procedures, may affect practices, the value provided to patients, and as a result, changes in purchasing volumes and pricing [24,25].

#### 2.2.2. Regulation and Strategies towards Cost-Reduction

Device costs for supply-intensive operations are intended to be impacted by bundled payments and gainsharing agreements. Some items are "*pass-throughs*", meaning international reimbursement programmes do not cover them. Examining these components' integration into the hospital and the effects of changing the incentives will help us to understand how policies are evaluated and developed. Early research on drug-eluting stents showed that gainsharing decreased expenses for coronary stent patients while maintaining quality and access, serving as a crucial example for academics hoping to comprehend the advantages of incentive-based programmes [26].

#### 2.2.3. Analyzing the Value and Comparative Effectiveness of Research

Assessing new products and product variety has become more methodical for healthcare organizations. However, studies on comparable products within a category and pricing transparency are scarce [17]. In that case, it may help to demonstrate product equivalencies, giving buyers the ability to enter the market with commitments to huge volumes and the associated leverage to negotiate lower prices [27]. Studies on value-based purchasing and purchasing innovation are very helpful for future health services research on the influence of evidence-based purchasing and its impact on cost [28,29].

#### 2.2.4. Preferences and Clinician Incentives

Physicians are surrogate buyers who use their professional experience and independence to choose items for hospitals and patients, particularly those in supply-intensive specialties like cardiology and orthopedics [30,31]. Physicians exercise great care in selecting the best medical devices and understand the significance of price as a selection factor. However, they also display little expertise and pay little attention to prices while choosing supplies [32]. Because of physicians' limited formal knowledge and information transparency regarding SCM, both during their training and once they are in practice, there is a lot of tension between the clinical domains and healthcare procurement [33]. Using AHA (American Hospital Association) survey data as a proxy, and assigning physicians based on their being in "*high-integration*" affiliated organizations, Grennan et al. (2022) considers the impact of physician integration on cost [13]. According to our study, such a proxy may only scrape the surface of the issue of encouraging physicians to think about the cost and procurement implications of their product choices in a high-integration system [34–36]. As mentioned, participating in gainsharing and other incentives may significantly impact physicians' use of expensive items.

### 2.3. Post-Pandemic Health Services Research and Supply Chain Perspective

In the COVID-19 era, every sector of the economy—from electronics and vehicles to agriculture, to imported goods and vaccinations, to everyday items like toilet paper—has at least one SC discussion. The same applies to healthcare. The "*ailing supply chain*" in healthcare has never been short of an opportunity or research questions [37]; SC concerns have risen to the top of the agendas and considerations of healthcare executives and politicians worldwide during the last two years. Expanding the emphasis beyond the SCs triple purpose of quality, cost, and outcomes to include SCRes (Supply Chain Resilience) and preparedness, as outlined below, in the mission statements of organizations in a sector with high resource-dependence, may be a starting point for change.

#### 2.3.1. Risk Assessment in Supply Chain

According to a company that handles and tracks SC risk for a variety of industries, disruptions from SCs were 67 percent higher in 2020 than in 2019 [38]. In the healthcare industry and other sectors, there is a paradigm change taking place, from a concentration on SC efficiency (i.e., cost-reduction) to one on SC robustness and contingency planning. Since SCs are now known to be more fragile than originally thought, other factors must take precedence above cost savings. Many medical devices in SCs are now vulnerable to disruptions as a result of the ubiquitous objectives in Six Sigma, Lean Management, and similar programmes that have decreased inventory and costs over the past two decades. Talks on safety stock and business continuity planning are currently taking place at all levels of organizational governance, from departmental units to the highest levels of the federal government. Recent ideas for the engagement of local bodies establishing pooled safety stockpiles continue to characterize the SNS' (Strategic National Stockpiles) role as a backup [39,40]. The health sector, which adopted a JIT (Just-In-Time) inventory and relied on suppliers and intermediates, must factor potential disruptions into their planning, readiness, and finance.

#### 2.3.2. Identifying the Risk of Resource-Dependency

The dependence on other businesses and the resource-dependence for healthcare products interact considerably. Medical gadgets reliant on semiconductors are expected to experience shortages and ensuing price increases. The *WSJ* (Wall Street Journal) highlighted the 2021 chip shortage for manufacturers of pacemakers, ultrasound equipment, and other devices [41]. In 2021, AdvaMed (Advanced Medical Technologies Association), which represents distributors of medical devices, urged the *DOC* (Department of Commerce) on the semiconductor industry to "*ensure that it does not cause SC disruption that affects healthcare delivery in the USA* [42]".

#### 2.3.3. Integration of the Supply Chain

The focus on integration in healthcare systems in the long run ignores a heavily researched and significant topic of SCM integration, which can be divided into integration with peers, intermediaries, suppliers, customers, or between organizational units [43,44]. In order to encourage innovation, clinical research, and the development of new services and processes, buyer–supplier integration with suppliers and intermediaries such as group purchasing organizations (GPOs) concentrates on partners that the hospital considers to be strategic [45]. When physician incentives influence supply selection decisions, for example, the physician–hospital integration could be viewed as a type of SC integration [35]. A strong case for horizontal SC integration to improve performance can usually be made with *decentralized health systems* [46] or recently merged systems. Nevertheless, numerous examples show that carrying out stated integration is difficult. SCM is among the most difficult areas to integrate. Years after the merger, numerous integrated systems continue to operate inefficient procurement processes (e.g., multiple supply information systems, separate procurement departments, multiple groups purchasing organizations, and overlapping contracts). Others have been able to integrate dispersed intermediaries by creating

pools and consolidated service centers [34]. Health Serv. Res. might significantly increase clarity in this crucial area.

### 2.4. Approaches to Examining Access and Justice in Healthcare/Healthcare Disparities

*Justice* and *health* are two of the most important issues being discussed in today's social and political discourse. Large differences in health and access to healthcare still exist, even in wealthy nations with highly established health insurance systems [47,48]. Additionally, the sharp enhancement in healthcare expenditures has led to growing social pressure to illustrate some rationing in publicly financed systems [49], and this rationing may further exacerbate unfair disparities or inequities. In developing countries where economic disparities are larger, and people have to pay a large portion of their healthcare costs out of their pocket [50], the issue of justice becomes even more pressing [51–55].

What justice needs in health and healthcare settings is deeply contested [56–60]. It is not easy to clearly distinguish "*justice*" from "*altruism*" or "*compassion*", but these are different concepts, and the distinction matters. A reference to justice creates stronger entitlements than "*compassion*" [61], which has implications for the macro-debate about how to organize the healthcare system and the micro-relations between healthcare professionals and patients with their families. When faced with suffering or death, people are presented with strong thoughts. The treatment of people in the context of health is more an issue of "*respect for human dignity*" than in most other areas of social organization. People might be profoundly injured in their self-respect and surprised via others' lack of respect if they feel they have been treated unfairly. Given this context, it is not surprising that "*justice and health*" have been significant research issues across a variety of academic disciplines [62,63].

#### 2.4.1. First Strategy: Philosophical Reflection

This strategy seeks to offer clear definitions of the importance of justice in health. Instances of such queries include: shall we be concerned about health or healthcare disparities, and if so, why (or why not)? What constitutes a "*need*" for healthcare? The specifics of the justice ideal have never been agreed upon and never will be. To come to a consensus is not the goal of philosophical reflection, though. Instead, it seeks to define the various ways that the idea of justice could be interpreted. Any meaningful scientific endeavor must start with conceptual clarity. A well-organized and cogent discussion of the rights and obligations of people in the area of health and healthcare also benefits from conceptual clarity. As a result, it plays a key role in every democratic decision-making process.

#### 2.4.2. Second Strategy: Based on Real-World Solutions

The second strategy of philosophical reflection is concerned with making conceptualizations of abstract justice operational and useful for understanding events in the real world, as opposed to the first method's focus on conceptualizations of abstract justice. For instance, reducing health disparities related to people's socioeconomic backgrounds can operationalize the abstract concept of justice and health. Under this operationalization, an empirical analysis of justice and health needs: (a) trustworthy insights into the effects of various policy options (for example, which policy options are most effective to decrease socioeconomic disparities in health?); (b) approaches for measuring such socioeconomic disparities in health; and (c) applications of these measurement approaches to the current situation (for example, how large are socioeconomic disparities in health in a given society at a given point? What fundamental factors give rise to this inequality?). Comparing the outcomes of empirical analyses based on various theories of justice is important for everyone, not just policymakers. The consequences of various justice concepts are sometimes best grasped when one is familiar with the policies they support [64].

#### 2.4.3. Third Strategy: Based on Lay Persons' Perceptions of Justice

The last strategy concentrates on how society as a whole perceives justice. Specific policy proposals are more likely to have an impact on actual social opinions than impersonal

conceptions of fairness. Instances of questions that employ the third tactic include: What do people believe about justice? How may we interpret their responses to particular policy proposals? When are allusions to justice just a ruse to conceal personal interests? How do the contrasts between self-interest, justice, and compassion that are important in a normative theory also apply to describing behavior? [65] The perceptions of less-educated persons and the cohesive concepts developed by philosophers may differ significantly. The latter are most important for comprehending how society functions because philosophical theories seldom ever adequately account for behavior. Focusing on real behavior raises more queries [66]. Are judgments of justice related to pure self-interest, and if so, how? How do they relate to other values and impulses like fraternity or compassion?

2.4.4. Fourth Strategy: Investigation into the Application of Justice Theories to Actual Policymaking

The fourth strategy analyzes perceptions of justice at the level of political decision-making during the examination of the third method—perceptions of justice at the individual level. Only with the fourth method can we determine whether, and if so, how much, reasoned ideas factor into the actual decision-making process. The second technique enables us to advise which policy options shall be used to realize a certain vision of justice. Numerous factors outside of normative theories are probably at work to explain, for instance, the significant difference in health insurance systems between nations. In the real world, how do ideas of justice interact with other social norms and financial restrictions? Does this variety represent cultural variations, such as variations in how people view justice? How does the political system influence people's attitudes toward justice, as examined using the third technique above? How does the structure of a political system affect the results of political decision-making? Each of these queries is pertinent on its own. There are, however, also clear feedback loops. Undoubtedly, the political system has an impact on social interactions and how citizens view justice. A CAN (coherent normative analysis) may naively provide unexpected and undesired results if the unique characteristics of the political institutions are not adequately taken into account [67].

## 3. Robust Optimization Protocol for General Integer and MLIP

Multiple works have been conducted on production schedules during the last few years. Most current works assume that each datum has known, fixed values. Nevertheless, uncertainty is prevalent in many scheduling problems [55] due to the absence of precise process models and variations in the procedure and environmental data. Therefore, new studies aimed to design techniques for addressing the scheduling problem based on uncertainty, to create reliable schedules that remained practicable in the exitance of the parameter uncertainty (refer to studies [68] as well as [68,69]). Hence, various methods could be utilized for the scheduling problem based on uncertainties like the probabilistic, stochastic, and fuzzy programs [70]. For comprehensive details on all formula-related information, please refer to Appendix A.

The study by [71] addressed the scheduling problem based on the demand's uncertainties. They employed a multi-stage stochastic MLIP pattern wherein several decisions were made by ignoring uncertainty. Moreover, other decisions have been made when uncertainty is identified. Therefore, Balasubramanian and Grossmann introduced an approximation approach to solving two-phase models based on the shrinking horizon strategy. Additionally, Ref. [72] utilized the concept of sensitivity analyses based on inferences for the MILP problem to determine the prominence of various limitations and factors in their scheduling system [73]. This system provided a series of candidate timetables for the unknown factors being considered.

Furthermore, Ref. [74] introduced a strategy for managing risks to scheduling with uncertain demands. They utilized a two-phase stochastic optimization pattern, maximizing the predicted profits and managing the risks explicitly via studying a novel aim as the measure for controls, which resulted in a multi-objective optimization model. Then, Ref. [75]

extended the above model to consider unknown processing time. Therefore, a two-phase stochastic procedure was employed wherein a weighted sum of the predicted makespan and predicted waiting for durations were minimized; then, the risks were measured with various accuracy measures. Finally, Ref. [76] dealt with the extension of the two-phase optimization model for examining possible accurate estimates of several unknown factors. Then, the split and bound strategy was applied to solve the issue according to a partition of the unknown area and an approximation of the bounds in the *Obj* function.

The robust optimization protocols have been designed for unknown information defined through multiple given distributions, like normal distribution, smooth dispersion, the difference of two NDs (normal distributions), BINDs (binomial dispersion), GDD (general discrete distribution), and Poisson dispersion. Our powerful optimization protocol introduced few auxiliary parameters and further limitations to the genuine MILP problem, which generated a deterministic strong counterpart problem, providing an optimal and possible solution for the relative volume of unknown information and the level of reliability, as well as feasibility tolerance. Consequently, a powerful optimization protocol was utilized for short-term scheduling issues based on uncertainties.

Now, a general ILP with unknown factors will be studied. However, there is a concern about developing a strong optimization protocol for generating "*reliable*" solutions to ILPs immune to data uncertainty. Hence, a generic ILP with $m \times n$ parameters and $m$ constraints would be considered:

$$\begin{cases} \text{Max} cx \\ s.t. Ax \le b, x \ge 0, \end{cases} \tag{1}$$

So that $A$ refers to an $m \times n$ integer matrix of rank $m$ and $b \in \Re^m$. Moreover, uncertainty results from the left side variables of inequality limitations $b_i, i = 1, 2, \ldots, m$. However, there are concerns about the feasibility of the limitations below in a robust optimization protocol framework;

$$\sum_{j \in J} a_{i,j} x_{i,j} \le b_i. \tag{2}$$

Based on [6], when the nominal data have been partly worried, one or more constraints could be substantially violated. Therefore, the current unit aimed to produce accurate solutions to a generic ILP problem that had immunity to uncertainty. Notably, our robust optimization protocol was initially provided via [6] for the LP problem, which has unknown coefficients, and consequently, Ref. [10] extended it for addressing an MILP problem. Notably, our method for introducing robustness into the original model was similar to the method employed in [10].

*3.1. Bounded Uncertainty*

Let us assume that uncertainty data have been in a range as in the following interval:

$$\left| \widetilde{a}_{i,j} - \overline{a}_{i,j} \right| \le \varepsilon \left| \overline{a}_{i,j} \right|, \left| \widetilde{b}_i - \overline{b}_i \right| \le \varepsilon \left| \overline{b}_i \right|, \tag{3}$$

where $\widetilde{a}_{i,j}, \widetilde{b}_i$ represent true values and $\overline{a}_{i,j}, \overline{b}_i$ refer to the nominal values (NVs). Moreover, $\varepsilon$ stands for the level of uncertainty.

The present study provided its description of the precise solutions to the ILP problem with the finite unknown left-hand side variables:

**Definition 1.** *When the uncertainty in ILP is illustrated in a bounded state, it is called solution x correct if it meets this condition:*

*(i)   x would it be possible for nominal problems.*

*(ii)   If we have true values (say, $\widetilde{b}_i$) of uncertain parameters from intervals (3). Therefore, x should meet i-th inequality constraints with the error of Max $\delta \max \left\{ 1, \left| \overline{b}_i \right| \right\}$, wherein $\delta$ would be explained as a certain level of infeasibility.*

*Particularly, condition (ii) could be written in this way:*

$$\forall i \left( \left| \widetilde{a}_{i,j} - \overline{a}_{i,j} \right| \le \varepsilon \left| \overline{a}_{i,j} \right|, \left| \widetilde{b}_i - \overline{b}_i \right| \le \varepsilon \left| \overline{b}_i \right|, \right):$$
$$\sum_{j \in J} \overline{a}_{i,j} x_{i,j} + \varepsilon \sum_{j \in M_J} \left| \overline{a}_{i,j} \right| u_{i,j} \le \widetilde{b}_i + \delta \cdot \max \left\{ 1, \left| \overline{b}_i \right| \right\}. \tag{4}$$

*Therefore, to derive a robust solution (RS), the worst values of the unknown variables have been used:*

$$\widetilde{b}_i \ge \overline{b}_i - \varepsilon \left| \overline{b}_i \right|, \tag{5}$$

*and we substituted (5) into (4).*

*Hence, x would be correct if and just if x was one of the possible solutions for the optimization problem below:*

$$\begin{aligned}
&\max \quad cx \\
&s.t. \sum_{j \in J} a_{i,j} x_{i,j} \le \overline{b}_i \\
&\sum_{j \in J} \overline{a}_{i,j} x_{i,j} + \varepsilon \sum_{j \in M_J} \left| \overline{a}_{i,j} \right| u_{i,j} \le \overline{b}_i - \varepsilon \left| \overline{b}_i \right| + \delta \cdot \max \left\{ 1, \left| \overline{b}_i \right| \right\} \\
&-u_{i,j} \le x_{i,j} \le u_{i,j} \\
&x_{i,j} \ge 0, \forall i, j.
\end{aligned} \tag{6}$$

*In addition, the calculated Formula (6) would be known as the "$(\varepsilon, \delta)$-Interval Robust Counterpart ($IRC[\varepsilon, \delta]$)" of the original ILP issue.*

*3.2. Symmetric Uncertainty*

Therefore, this sub-section supposed uncertain data $\widetilde{a}_{i,j}, \widetilde{b}_i$ as a random and symmetric distribution surrounding the NVs $\overline{a}_{i,j}, \overline{b}_i$ as follows:

$$\widetilde{a}_{i,j} = (1 + \varepsilon \xi_{i,j}) \overline{a}_{i,j}, \widetilde{b}_i = (1 + \varepsilon \varsigma_i) \overline{b}_i, \tag{7}$$

where the perturbations $\xi_{i,j}, \varsigma_i$ represent the independent variables with symmetric distribution in the interval $[-1, 1]$.

To provide the same description of the correct solution to the ILP problem with finite uncertainty, transferring a deterministic version (ii) to the common probabilistic one would be very crucial. Hence, we defined a correct solution to the ILP problem with the symmetric unknown variables:

**Definition 2.** *When there is a symmetric uncertainty, the solution x would be correct if it meets this condition:*

*(i)*    *x would be possible for the nominal problems.*
*(ii)*   *For each i event probability of a limited violation; that is,*

$$\sum_{j \in J} \widetilde{a}_{i,j} x_{i,j} > \widetilde{b}_i + \delta \cdot \max \left\{ 1, \left| \overline{b}_i \right| \right\},$$

*It would be, at most $\kappa$, that $\delta > 0$ refers to a certain impossible tolerance and $\kappa > 0$ refers to a certain level of reliability.*

*Hence, a correct situation to the ILP (integer linear programming) problem with the symmetric unknown parameters would be calculated via making solutions to the ($\varepsilon, \delta, \kappa$)-robust counterpart (RC [$\varepsilon, \delta, \kappa$]):*

$$
\begin{aligned}
&\max cx \\
&s.t. \sum_{j \in J} \overline{a}_{i,j} x_{i,j} \leq \overline{b}_i \\
&\sum_{j \in J} \overline{a}_{i,j} x_{i,j} + \varepsilon \left[ \sum_{j \in M_J} \left| \overline{a}_{i,j} \right| u_{i,j} + \Omega \sqrt{\sum_{j \in M_J} \overline{a}_{i,j}^2 v_{i,j}^2 + \overline{b}_i^2} \right] \leq \overline{b}_i + \delta \max\{1, |\overline{b}_i|\} \\
&-u_{i,j} \leq x_{i,j} - v_{i,j} \leq u_{i,j}, \forall i, j, \\
&x_{i,j} \geq 0, \forall i, j,
\end{aligned}
\tag{8}
$$

*where $\Omega$ stands for a (+) parameter via $\kappa = \exp\left\{-\Omega^2/2\right\}$. Therefore, the case considered here is a specific kind of study conducted by [10]; hence, the procedure for deriving RC (robust counterpart) [$\varepsilon, \delta, \kappa$] could be immediately called **Lemma 1** and **Theorem** 2. These were reported by [10] and have been ignored in the present research for simplicity.*

*Notably, according to the above explanation of the relatively correct formulation, the relative variables of uncertainties ($\varepsilon$), reliability level ($\kappa$), and infeasibility tolerance ($\delta$) have been supposed to be single and common for simplification. Nonetheless, these new robust optimization protocols could be readily extended for considering more general cases where such parameters depended on the constraints.*

## 4. Site Selection Model for QEH

The site selection model is concerned with choosing the right site for newly opened regional hospitals to maximize utilization. The involved model is related to three steps:

➢ Develop a utilization matrix;
➢ Specify constraints;
➢ Apply a robust optimization protocol approach to select the best site.

### 4.1. Utilization Matrix

Estimating each choice's potential utilization or "*attractiveness*" is necessary to optimally locate a hospital throughout a region. The mentioned utilization or "*attractiveness*" may be measured by the number of subscribers who select this site location under a dual-choice option.

We explored the probability of the site selection as a function of travel time between the patient's residence and hospital location and socioeconomic attributes. The utilization function gives the expected utilization $u_{i,j}$ from population unit $i$ to facility location $j$ for each $i, j$ combination:

$$
u_{i,j} = n_i p_{i,j},
\tag{9}
$$

where $p_{i,j}$ is the probability of utilization for a person from a population unit $i$ location $j$ as determined from the utilization function, and $n_i$ is the total population of the unit $i$. The utilization matrix also gives the Max expected utilization for each choice:

$$
u_j = \sum_{i=1}^{m} u_{i,j},
\tag{10}
$$

### 4.2. Constraints

Constraints are imposed on the model to limit expenses, ensure that hospitals exceed a Min utilization, assign each population unit to one and only one hospital, and restrict the number of hospitals.

The cost constraint could be considered in terms of fixed and variable costs. Fixed costs may represent set-up expenditures, including planning, capital outlays for equipment

and facilities, and recruiting the necessary core staff. Variable costs are those expenditures that are a function of the number of subscribers.

$$\sum_{j=1}^{n} f_j y_j + \sum_{j=1}^{n} v_j (\sum_{i=1}^{m} u_{i,j}) x_{i,j} \leq C, \tag{11}$$

where $f_j$ and $v_j$ are the fixed and variable costs, respectively, and $C$ represents the total budget amount. The variables $y_j$ are 1 when a hospital is to be located at the site $j$, and are 0 otherwise.

It is believed that there exists a break-even point or Min utilization for a hospital.

$$\sum_{i=1}^{m} u_{i,j} x_{i,j} \geq k y_i, \tag{12}$$

So that $k$ is the Min expected enrollment requirement before a hospital would be opened at the site $j$.

Each population unit would choose one and only one hospital, which could be expressed numerically as follows:

$$\sum_{j=1}^{n} x_{i,j} \geq 1. \tag{13}$$

Furthermore, the Max number of hospitals should be prescribed by HA managers,

$$\sum_{j=1}^{n} y_j \leq S. \tag{14}$$

Finally, the mathematical model to select the site to determine the location of a new hospital that maximizes total utilization is presented below:

$$\begin{aligned}
\text{Maximize } E = & \sum_{i=1}^{m} \sum_{j=1}^{n} u_{i,j} x_{i,j} \\
\text{s.t. } & \sum_{j=1}^{n} f_j y_j + \sum_{j=1}^{n} v_j (\sum_{i=1}^{m} u_{i,j}) x_{i,j} \leq C, \\
& \sum_{i=1}^{m} u_{i,j} x_{i,j} \geq \text{k} y_i, \\
& \sum_{j=1}^{n} x_{i,j} \geq 1, \\
& \sum_{j=1}^{n} y_j \leq S.
\end{aligned} \tag{15}$$

## 5. Robust Optimization

It is possible to experience a direct application of the robust optimization protocol models in Section 2 to the site selection problem with uncertain costs and budgets. Remembering the basic model (15) $IRC[\varepsilon, \delta]$ and $RC[\varepsilon, \delta, \kappa]$, its application to this basic formulation has been considered. Considering the context of hotel revenue management, parameters $\bar{b}_i$ in (6) and (8) would surely be positive. Therefore, it would be completely reasonable to assume the existence of less than one booking request arriving every day. Hence, $\max\{1, |\bar{b}_i|\} = \bar{b}_i$.

Consequently, the suggested robust site selection models can be converted into the following formulations:

$$
\text{Maximize } E = \sum_{i=1}^{m} \sum_{j=1}^{n} u_{i,j} x_{i,j}
$$

$$
\text{s.t. } \sum_{j=1}^{n} \overline{f}_j y_j + \sum_{j=1}^{n} \overline{v}_j (\sum_{i=1}^{m} u_{i,j}) x_{i,j} \leq \overline{C},
$$

$$
\sum_{j=1}^{n} \overline{f}_j y_j + \varepsilon \sum_{j \in M_J} \left| \overline{f}_j \right| s_j + \sum_{j \notin K_J} \overline{v}_j (\sum_{i=1}^{m} u_{i,j}) x_{i,j} +
$$

$$
\sum_{j \in K_J} ((\overline{v}_j + \varepsilon \left| \overline{v}_j \right|)(\sum_{i=1}^{m} u_{i,j}) x_{i,j}) \leq (1 - \varepsilon + \delta)\overline{C}, \tag{16}
$$

$$
-s_j \leq y_j \leq s_j
$$

$$
\sum_{i=1}^{m} u_{i,j} x_{i,j} \geq \text{ky}_i,
$$

$$
\sum_{j=1}^{n} x_{i,j} \geq 1,
$$

$$
\sum_{j=1}^{n} y_j \leq S.
$$

and

$$
\text{Maximize } E = \sum_{i=1}^{m} \sum_{j=1}^{n} u_{i,j} x_{i,j}
$$

$$
\text{s.t. } \sum_{j=1}^{n} \overline{f}_j y_j + \sum_{j=1}^{n} \overline{v}_j (\sum_{i=1}^{m} u_{i,j}) x_{i,j} \leq \overline{C},
$$

$$
\sum_{j=1}^{n} \overline{f}_j y_j + \sum_{j=1}^{n} \overline{v}_j (\sum_{i=1}^{m} u_{i,j}) x_{i,j} +
$$

$$
\varepsilon \left[ \sum_{j=1}^{n} \left| \overline{f}_j \right| l_j + \Omega \sqrt{ \sum_{j \in M_J} \overline{f}_j^2 z_j^2 + \sum_{j \in M_J} \overline{v}_j^2 z_j^2 + \overline{C}^2 } \right] \leq (1 + \delta)\overline{C}, \tag{17}
$$

$$
\sum_{i=1}^{m} u_{i,j} x_{i,j} \geq \text{ky}_i,
$$

$$
\sum_{j=1}^{n} x_{i,j} \geq 1,
$$

$$
\sum_{j=1}^{n} y_j \leq S.
$$

### 5.1. Robust Optimization Protocol to Schedule under Uncertainty

According to the research design, a robust optimization protocol formulation (for more information about protocol formulation, see our other paper—Heydari, M. et al., 2021) has been utilized for four instance problems. Each instance has been run through GAMS software 28.2.0v (GAMS Software GmbH, New York, NY, USA) [77] on a 3.20 GHz Linux workstation. Then, CPLEX 8.1 was utilized to solve the MILP problems, whereas DICOPT was utilized to solve the MINLP problems [78].

#### 5.1.1. Instance 1. Uncertainties via a Poisson Distribution during Processing Time

Ref. [79] initially designed an instance process utilized as a motivating instance in Section 1 of the article [10] on finite uncertainties. Therefore, two products have been generated through three feeds following the State-Task Network (Figure 1). Then, the STN (State-Task Network) utilized three kinds of tasks that could be performed in four diverse units. Table 1 reports the corresponding data for this instance, including suitability, capacity, processing time, and storage limitation. It aimed at maximizing the profit from selling the products fabricated in the timetable in twelve hours.

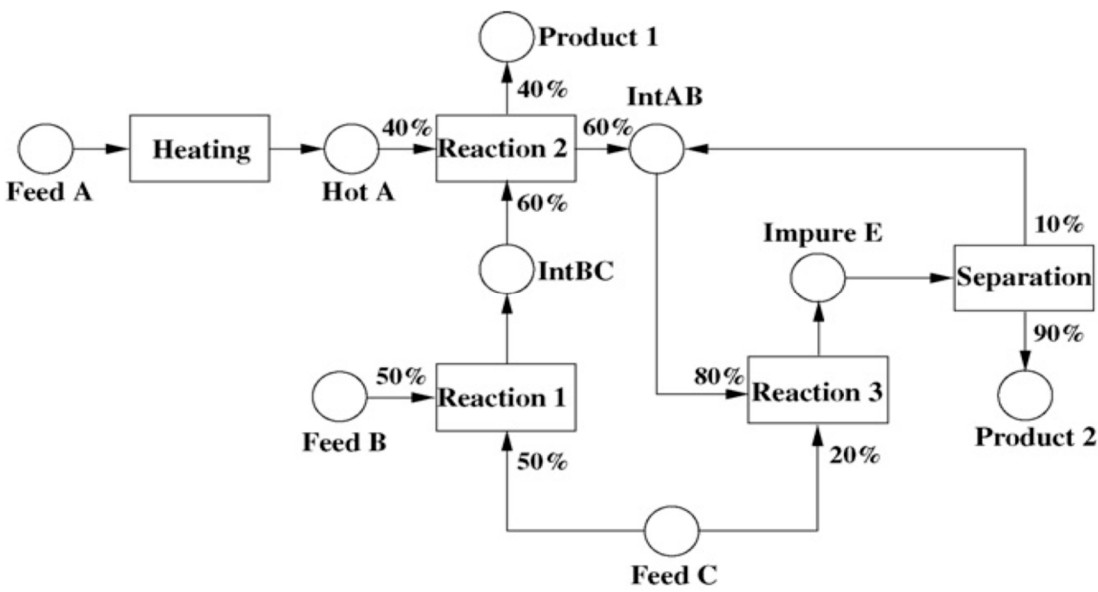

**Figure 1.** The state-task network. For instance, 1.

**Table 1.** The data. For instance, 1.

| Units | Adaptability | Process Period | Capacity |
|---|---|---|---|
| Heater | Heating | 1.0 | 100 |
| Reactor 1 | R 1, 2, 3 | 2.0, 2.0, 1.0 | 50 |
| Reactor 2 | Reactions 1, 2, 3 | 2.0, 2.0, 1.0 | 80 |
| Separator | Separation | 2.0 | 200 |
| States | Initial Amount | Price | Storage |
| Feed A | Unbounded | 0 | Unbounded |
| Feed B | Unbounded | 0 | Unbounded |
| Feed C | Unbounded | 0 | Unbounded |
| Hot A | 0 | 0 | 100 |
| IntAB | 0 | 0 | 200 |
| IntBC | 0 | 0 | 150 |
| ImpureE | 0 | 0 | 200 |
| Product 1 | 0 | 10.0 | Unbounded |
| Product 2 | 0 | 10.0 | Unbounded |

Suppose that uncertainties during the processing time would have a Poisson distribution with a value equal to 5, a level of uncertainty ($\in$) equal to 5%, infeasibility tolerance ($\delta$) equal to 20%, and a reliability level ($\kappa$) equal to 24% (relative to $\lambda$-value equal to 6). When the *RC* [$\in$, $\delta$, $\kappa$] problem has been solved, a "*robust*" schedule has been achieved (Figure 2) that investigates uncertainty in the processing time. Figure 2 depicts the nominal schedule.

Compared with the nominal solution (NS) achieved at the NVs of the processing duration, a correct situation exhibited many distinct scheduling approaches. For instance, even the sequence of tasks in the two reactors in Figure 3 experienced a significant deviation from Figure 1.

Moreover, compared with the NS attained at the NVs of the processing time, a correct situation was exhibited, so various scheduling approaches were used. For instance, even the task sequences in the two reactors in Figure 3 considerably deviated.

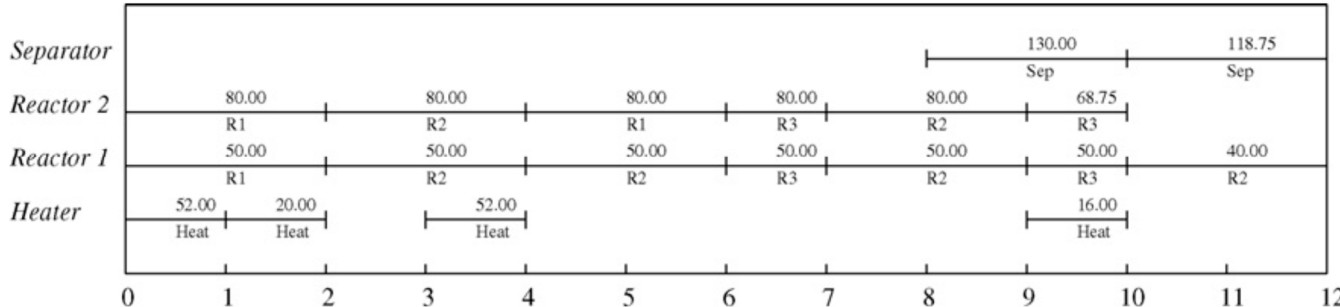

**Figure 2.** An optimal solution via the nominal time for processing (profit = 3638.75).

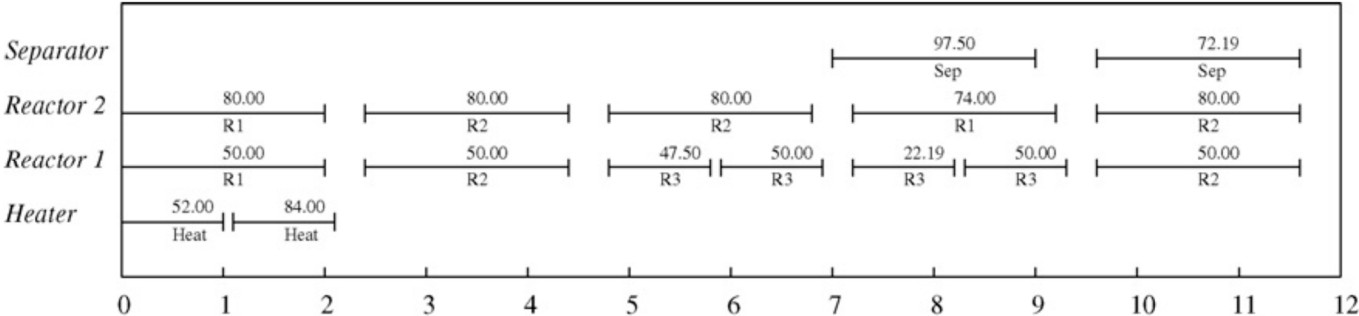

**Figure 3.** An RS via uncertain processing time (profit = 2887.19).

The sequences in NS are shown in Figure 2. As seen, the correct solutions ensured the feasibility of a correct schedule with a given level of uncertainty, reliability, and infeasibility tolerance. Nonetheless, a reduction in the resultant profit from USD 3638.75 to 2887.19 has been reported, representing the effects of uncertainty on the overall production. Table 2 compares the pattern and solution statistics for robust and NSs.

**Table 2.** The model and solution statistics. For instance, 1.

|  | RS | NS |
|---|---|---|
| Benefit | 2887.19 | 3638.75 |
| CPU time (s) | 11.33 | 0.46 |
| BIN variables | 96 | 96 |
| CON variables | 442 | 442 |
| Constraints | 777 | 553 |

Figure 4 is a summary of the outputs of the RC problem with numerous diverse mixes of infeasibility and uncertainty levels at increasing values of the level of reliability. As shown, at a certain level of reliability, the maximal profit that could be gained decreased by increasing the level of uncertainty, reflecting further conservative scheduling decisions due to uncertainty. Moreover, at the specific level of reliability, maximal profit increased by enhancing the tolerance level of infeasibility. It is possible to incorporate a more aggressive scheduling arrangement if violations of the pertinent timing constraints could be further tolerated. Additionally, at the specific level of uncertainty and infeasibility tolerance, profit increased by enhancing the level of reliability, reflecting that probable violation of uncertain constraints allowed for more aggressive scheduling. Therefore, the obtained outputs would be compatible with intuition and other approaches. Nevertheless, considering the powerful optimization protocol, the impacts of uncertainties and the trade-off between the opposed goals would be effectively and rigorously quantified.

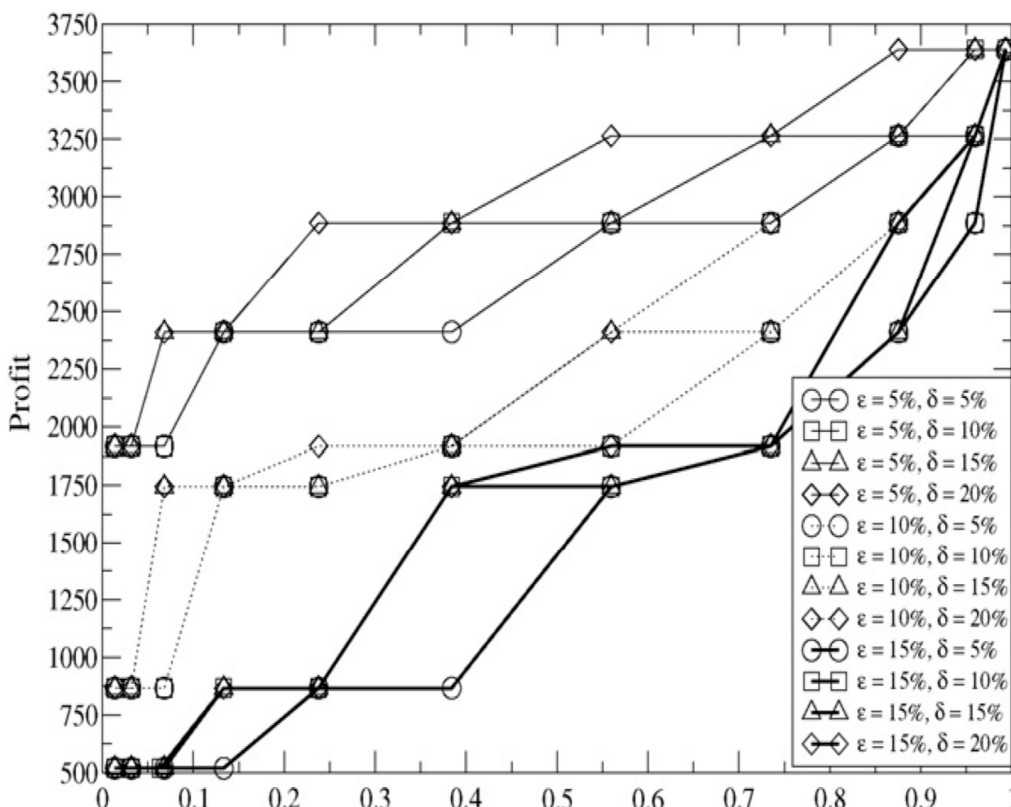

**Figure 4.** Level of profit versus reliability.

### 5.1.2. Instance 2. Uncertainty via a Smooth Distribution in the Demand of the Goods

According to instance 2, uncertainties have been considered with a smooth dispersion in demand for the goods for a similar procedure provided in instance 1. Nonetheless, the objective function has been to minimize the makespan for a certain demand equal to 70 for good 1 and 80 for good 2. The level of uncertainties ($\in$) equaled 10%, and infeasibility tolerance ($\delta$) equaled 5%. Moreover, the level of reliability ($\kappa$) was 0%. Figure 5 demonstrates a nominal schedule with a makespan equal to 8.007. According to Figure 6, a robust schedule was achieved by solving the robust counterpart problem so that the corresponding makespan equaled 8.174. In the case of the execution of the obtained schedule, the makespan would be at most 8.174 with a 100% probability with the existence of 10% uncertainty in product demand. Table 3 compares the pattern and solution statistics for robust and NS site selection [80].

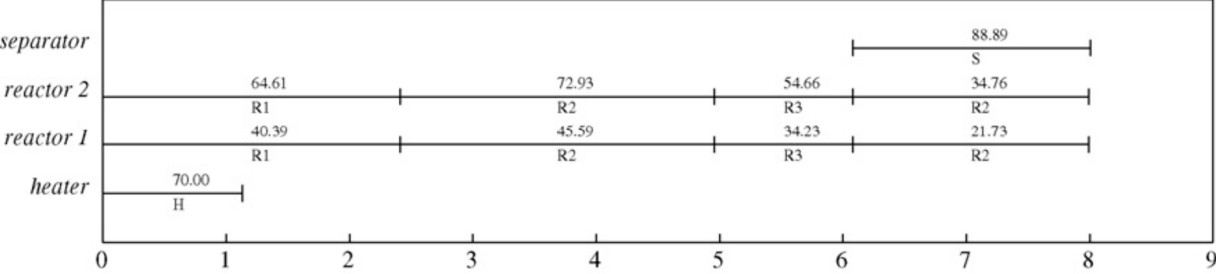

**Figure 5.** Optimum solution via nominal goods demands (makespan = 8.007).

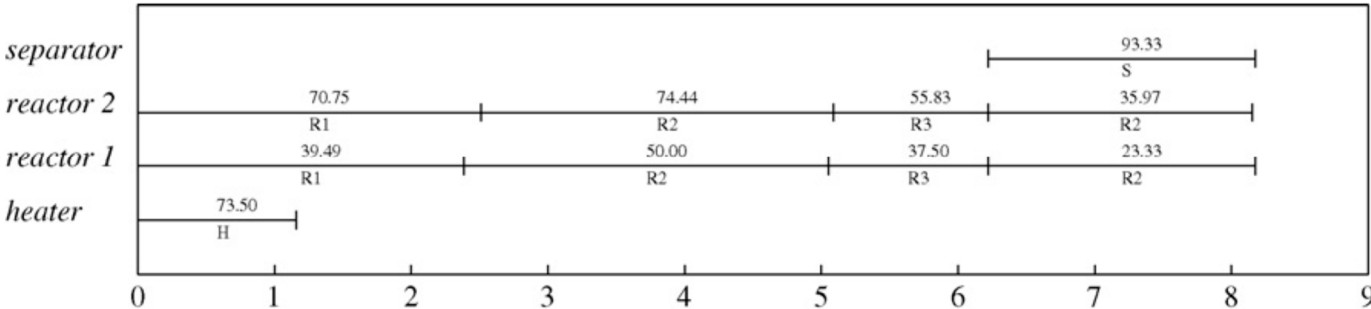

**Figure 6.** An RS via unknown goods demand (makespan = 8.174).

**Table 3.** The model and solution statistics. For instance, 2.

|  | RS | NS |
|---|---|---|
| Makespan | 8.174 | 8.007 |
| CPU time (s) | 0.02 | 0.02 |
| BIN variables (Binary variables) | 60 | 60 |
| CON variables (Continuous variables) | 280 | 280 |
| Constraints | 409 | 375 |

Figure 7 represents a summary of the outputs of the RC problem with multiple diverse combinations of infeasibility as well as uncertainty levels at the enhancing values of the level of reliability. As observed, at a certain level of reliability, the Min makespan increased by enhancing the level of uncertainty, indicating more conservative scheduling decisions, which further lasted due to uncertainties in demand. Moreover, at the constant levels of reliability, the Min makespan decreased by increasing the level of infeasibility tolerance, meaning that scheduling arrangements with higher aggression could be included if a violation of the relevant demand limitations.

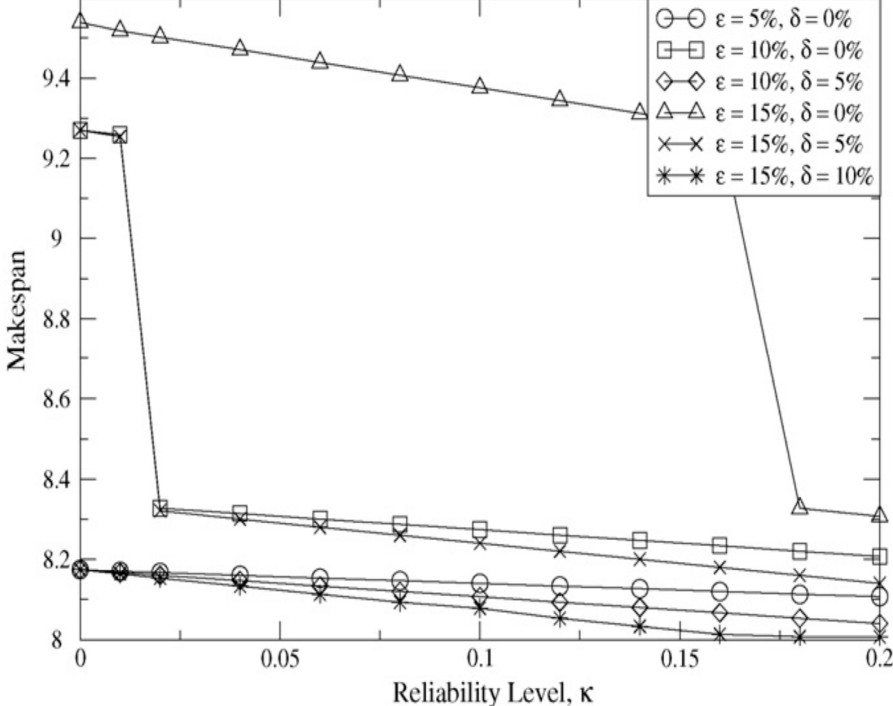

**Figure 7.** The makespan versus the level of reliability at diverse levels of uncertainties as well as infeasibility level. For instance, 2.

It could be further allowed. Put differently, at a certain levels of uncertainty and infeasibility tolerance, makespan decreased by increasing reliability, reflecting that probable violations of uncertain constraints allowed for more aggressive scheduling. Hence, it is possible to quantify the effects of uncertainty on scheduling using a robust optimization protocol.

### 5.1.3. Instance 3. Uncertainty via Normal Distribution in the Market Price

For this instance, the level of uncertainty has been investigated via a normalized distribution in the market price for a procedure similar to that used in instances 1 and 2. Nonetheless, the objective function was to maximize profits in eight hours. The levels of uncertainty ($\in$), infeasibility tolerance ($\delta$), and level of reliability ($\kappa$) equaled 5, 5, and 5%. Figure 8 shows a nominal schedule with a profit equal to 1088.75. Moreover, the robust schedule was achieved via solving the robust counterpart issue (Figure 9), and the corresponding advantage equaled 966.97. Upon the implementation of the schedule, the profit was ensured to not be less than 966.97, with a 95% probability, with the existence of 5% uncertainty in the raw materials and product prices. Table 4 compares the pattern and solution statistics for robust and NS site selection.

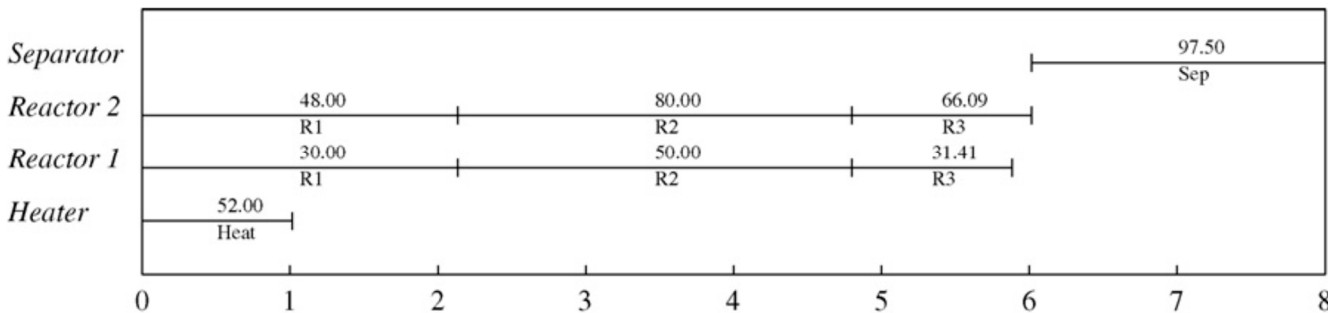

**Figure 8.** The optimized solutions via the nominal market price (profits = 1088.75).

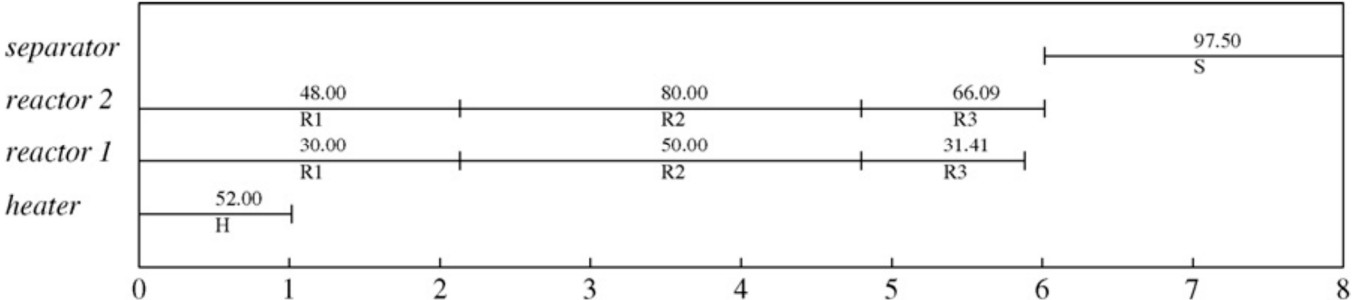

**Figure 9.** A correct solution via unknown market price (profits = 966.97).

**Table 4.** The model and solution statistic. For instance, 3.

|  | RS | NS |
|---|---|---|
| Benefit | 966.97 | 1088.75 |
| CPU time (s) | 0.05 | 0.02 |
| BIN variables | 60 | 60 |
| CON variables | 280 | 280 |
| Constraints | 334 | 334 |

Figure 10 represents a summary of RC problem outputs at multiple diverse levels of uncertainties and 0% infeasibility tolerances at the enhancing amounts of levels of reliability. As seen in the figure, at a certain level of reliability, the maximal profit that could be gained decreased by enhancing the level of uncertainty, indicating more conservative scheduling

decisions due to uncertainty. Moreover, at a certain level of uncertainty and infeasibility tolerance, profit increased by enhancing the level of reliability, demonstrating that with the increase in probable violation of uncertain constraints or κ, λ decreased, and the profit took on a greater value based on Equation (18):

$$Profit \leq \sum_{s \in S_p} p_s . STF(s) - \in \lambda \sqrt{\sum_{s \in S_p} p_s^2 STF(s)^2 + \sum_{s \in S_r} p_s^2 STI(s)^2 + \delta} \qquad (18)$$

**Figure 10.** For instance, the profit versus the reliability level at diverse uncertainties and infeasibility levels, 3.

So that $\lambda = F^{-1}(1-k)$ and $F_n^{-1}$ are the reverse distribution functions of the random variables with the standard normalized distribution.

Notably, the above instance considered uncertainties during the procedure time of missions for the industrial empirical report initially provided in [81]. Therefore, actual plant data have been utilized for determining the kinds and levels of uncertainty in the processing time. The industrial plant was a multi-product chemical plant, which manufactured tens of various goods following a major three-phase recipe and its changes with ten pieces of instrumentation. Therefore, the first sub-horizon in [81] study was 5 days and eight products. Hence, the objective function was to maximize the general production, described as the weighted sum of the substances aggregated after the sub-horizon minus a penalty term for the lack of satisfaction of the demands with mid-term deadlines. Finally, the processing recipe demonstrated in Figure 11 was utilized for each product.

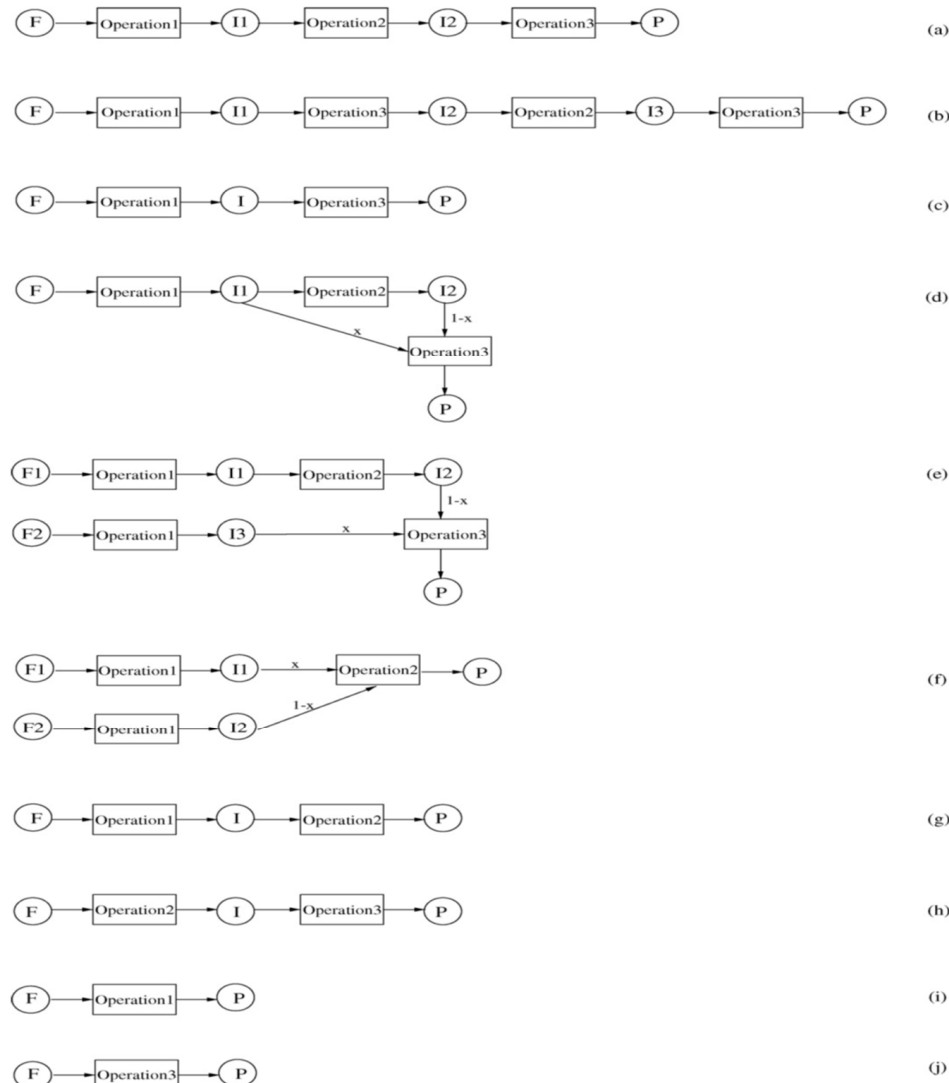

**Figure 11.** The state-mission network of the production recipe in the industrial empirical report (**a**–**j**).

This plant consisted of three kinds of units, each corresponding to one of three major processing operations. Therefore, four types of one unit (units 1 to 4) were utilized for operation 1, three types of two units (units 5 to 7) were employed for operation 2; and three types of three units (units 8 to 10) were utilized for operation 3. Then, type 1 and type 3 units were applied in the batch mode, whereas type 2 units acted in a continual mode. Moreover, Table 5 presents the nominal processing duration or the processing rates of all tasks in the relative proper units.

Therefore, to determine the forms of uncertainties in the processing duration or rate, we addressed the analysis of the actual plant data. Hence, two kinds of uncertainty were selected based on the data: uncertainties via a normal distribution and finite uncertainties. For bounded uncertainty, ranges for unknown variables have been provided, and mean and standard deviation (SD) for uncertain parameters have been determined for normal uncertainty. Moreover, a total number of 23 uncertain parameters was recognized, including 8 in units 1 to 4; 5 in units 5 to 7; and 10 in units 8 to 10. Table 6 summarizes all uncertain parameters' characteristic NVs, means, ranges, and SDs.

**Table 5.** The nominal processing rates and times in industrial evidence.

| Mission | Units | | | | | | | | | |
|---|---|---|---|---|---|---|---|---|---|---|
| | 1 | 2 | 3 | 4 | 5 | 6 | 7 | 8 | 9 | 10 |
| 1 | 0 | 0 | 0 | 9.5 | -- | -- | -- | -- | -- | -- |
| 2 | -- | -- | -- | -- | 0 | 0 | 0.95 | -- | -- | -- |
| 3 | -- | -- | -- | -- | -- | -- | -- | 12 | 12.8 | 12.5 |
| 4 | 0 | 10 | 10 | 10 | -- | -- | -- | -- | -- | -- |
| 5 | -- | -- | -- | -- | 0.575 | 0.575 | 0.725 | -- | -- | -- |
| 6 | -- | -- | -- | -- | -- | -- | -- | 12 | 12.8 | 12.5 |
| 7 | 6.09 | 6.09 | 6.09 | 11.1 | -- | -- | -- | -- | -- | -- |
| 8 | -- | -- | -- | -- | 0.6 | 0.6 | 0.8 | -- | -- | -- |
| 9 | -- | -- | -- | -- | -- | -- | -- | 12.5 | 13.8 | 12.9 |
| 10 | 6.09 | 6.09 | 6.09 | 11.1 | -- | -- | -- | -- | -- | -- |
| 11 | -- | -- | -- | -- | 0.6 | 0.6 | 0.8 | -- | -- | -- |
| 12 | -- | -- | -- | -- | -- | -- | -- | 12.5 | 13.8 | 12.9 |
| 13 | 6.09 | 6.09 | 6.09 | 11.1 | 0.6 | 0.6 | 0.8 | -- | -- | -- |
| 14 | -- | -- | -- | -- | -- | -- | -- | -- | -- | -- |
| 15 | -- | -- | -- | -- | -- | -- | -- | 12.5 | 13.8 | 12.9 |
| 16 | -- | -- | -- | -- | 0.6 | 0.6 | 0.8 | -- | -- | -- |
| 17 | -- | -- | -- | -- | -- | -- | -- | 12.5 | 13.8 | 12.9 |
| 18 | 0 | 8.5 | 8.5 | 0 | -- | -- | -- | -- | -- | -- |
| 19 | -- | -- | -- | -- | -- | -- | -- | 0 | 15 | 16 |
| 20 | 0 | 0 | 8.38 | 9.5 | -- | -- | -- | -- | -- | -- |

**Table 6.** The bounded and normalized uncertainties in the processing duration and rate for the empirical report.

| Mission | Unit | NVs | Uncertainty | Rang | Mean | SD |
|---|---|---|---|---|---|---|
| 1 | 4 | 9.5 | N | -- | 9.912 | 0.523 |
| 7,10,13 | 1–3 | 6.09 | N | -- | 6.153 | 0.152 |
| 7,10,13 | 4 | 11.1 | B | 10.1–11.3 | -- | -- |
| 20 | 3 | 8.38 | B | 8.00–10.42 | -- | -- |
| 2 | 7 | 0.95 | N | -- | 0.9611 | 0.112 |
| 8,11,14,16 | 5–6 | 0.60 | B | 0.344–0.853 | -- | -- |
| 3,6 | 9 | 12.8 | B | 10.5–19.3 | -- | -- |
| 9,12,15,17 | 9 | 13.8 | B | 12.0–16.3 | -- | -- |
| 9,12,15,17 | 10 | 12.9 | N | -- | 12.100 | 0.760 |

Normal = N; Bounded = B.

Strategy 2 for uncertainties in the processing duration or the rate in Section 3.2 were utilized for the mentioned case study. Besides major sequencing constraints, the processing times appeared in two further constraints associated with timing of operation 1's mission:

$$T^s(i, j, n+1) \leq T^f(i, j, n) + H(2 - wv(i, j, n) - wv(i, j, n+1))$$
$$\forall_i \in I_r, j \in J_i, n \in N, n \neq N,$$
$$T^s(i, j, n+1) \leq T^f(i, j, n) + tcl_{ii'} + H\left(2 - wv(i, j, n) - wv\left(i', j, n+1\right)\right) \quad (19)$$
$$\forall_i \in I_r, i, i' \in I_j, i \neq i', n \in N, n \neq N$$

So that $I_r$ represents a series of operation 1 tasks. $J_r$ refers to a collection of type 1 units appropriate for operation 1 tasks. When $T^f(i, j, n)$ variables are substituted, the further constraints proposed below for parameters via the bounded uncertainty are for obtaining the robust counterpart issue:

$$T^s(i, j, n+1) - T^f(i, j, n) \leq a_{ij}^L.wv(i, j, n) +$$
$$\beta_{ij}^L.B(i, j, n) + H(2 - wv(i, j, n) - wv(i, j, n+1)) + \delta 2, \quad (20)$$

$$T^s(i, j, n+1) - T^s\left(i', j, n\right) \le a_{ij}^L . wv\left(i', j, n\right) +$$
$$\beta_{ij}^L . B\left(i', j, n\right) + tcl_{ii'} + H(2 - wv(i, j, n) - wv\left(i', j, n+1\right)) + \delta 2, \tag{21}$$

Here, $a_{ij}^L = (1- \in)a_{ij}$, $\beta_{ij}^L = (1- \in)\beta_{ij}$, and $\delta 2$ represent a varied and correlated relationship, as they are followed with factor δ, which participated in further limitations relative to major sequencing limitations:

$$\delta + \delta 2 = a^U - a^L = 2. \in .a,$$
$$or\ \delta + \delta 2 = \beta^U - \beta^L = 2. \in .\beta. \tag{22}$$

Accordingly, further limitations would be proposed for variables with normal uncertainties:

$$T^s(i, j, n+1) - T^s(i, j, n) \le \left[1- \in \left(\lambda_{ij}^a . \sqrt{\sigma_{ij}^a} - \mu_{ij}^a\right)\right] . a_{ij} . wv(i, j, n)$$
$$+ \left[1- \in \left(\lambda_{ij}^\beta . \sqrt{\sigma_{ij}^\beta} - \mu_{ij}^\beta\right)\right] . \beta_{ij} . \beta(i, j, n) \tag{23}$$
$$+ H(2 - wv(i, j, n) - wv(i, j, n+1)) + \delta 2,$$

$$T^s(i, j, n+1) - T^s\left(i', j, n\right) \le \left[1- \in \left(\lambda_{i'j}^a . \sqrt{\sigma_{i'j}^a} - \mu_{ij}^a\right)\right] . a_{i'j} . wv\left(i', j, n\right)$$
$$+ \left[1- \in \left(\lambda_{i'j}^\beta . \sqrt{\sigma_{i'j}^\beta} - \mu_{i'j}^\beta\right)\right] . \beta_{i'j} . \beta\left(i', j, n\right) + tcl_{i'i} \tag{24}$$
$$+ H(2 - wv(i, j, n) - wv\left(i', j, n+1\right)) + \delta 2$$

So that $\delta 2$ would be described as:

$$\delta + \delta 2 = 2. \in \left(\lambda . \sqrt{\delta} - \mu\right). \tag{25}$$

As seen, the objective function for the above issue would be to maximize the production of the relative values of each state minus a penalty term for the lack of contentment of the demand at the intermediate due date:

$$\gamma \sum_s vald_s valp_s valm_s STF(s) - \sum_s \sum_n pri_{sn} SL(s, n)$$
$$\forall_s \in S,\ n \in N \tag{26}$$

So that $vald_s$ represent the corresponding values of the relative product reflecting the respective significance for fulfilling the future demands. In addition, $valp_s$ refers to the relative value of the corresponding products representing the respective priority, and $valm_s$ stands for the relative values of the state (s) in the materials sequences for the corresponding products. Moreover, STF(s) indicates the amounts of state (s) at the end of the horizon, and $pri_{sn}$ refers to the demand priority (making choices about which treatments are covered by insurance and which are not is referred to as prioritizing, or, negatively, as rationing). All insurance systems require these judgments, but publicly supported government systems are the most challenging. Cost-effectiveness analysis (CEA), which aims to "produce" the greatest number of quality-adjusted life years with a given government budget, is the dominant technique for making these judgments. This conventional method prioritizes effectiveness above justice. However, this is the very reason it poses challenging justice questions. Individual patients will have uneven access to healthcare if they can pay for non-reimbursed therapies out of pocket. The same holds true in a less-severe scenario when patients are required to make significant co-payments for treatments that are only partially covered. The problem presented by uncommon diseases is particular. The so-called orphan pharmaceuticals, or medications for rare diseases, are rarely covered by insurance, and without government funding, the research and development of these medications is typically not profitable. However, it is challenging to argue that people with rare genetic disorders should not be treated just because they are few from an ethical standpoint [82].

for the state (s) at the event point (n). Furthermore, SL (s, n) represents a slack variable for the number of states (s) that have not met the demand at the event point (n), and γ stands for a fixed coefficient applied for balancing the relative value of two terms in the *Obj* function.

It should be noted that this problem required additional sequencing constraints ((19) to (21)) for the accurate scheduling of the operation 1 task. However, using such constraints to account for problem uncertainty led to a MILP problem because they only had one uncertain parameter, which produced linear deterministic types for typical unknown and constrained restrictions.

### 5.2. Computational Outputs and Discussion

Figure 12 depicts an NS for the problem with continual time formula and objective function values equal to 121.37. Figure 13 presents a solution to the correct respective problem, with each of the 23 uncertain parameters at 10% (relative) infeasibility tolerances (δ) for the finite unknown variables and 20% for normal uncertain parameters; a 5% level of uncertainty (∈); and a 5% level of reliability (κ) for normal uncertain parameters, reflecting just 5% violation of the constraints. As seen, the *Obj* function value is 105.76, and the processing time of all uncertain tasks was extended to ensure that the schedule would be feasible at the given levels of uncertainty, reliability, and infeasibility tolerances. Nonetheless, the value of the *Obj* function declined. Moreover, a more precise investigation of the term related to the *Obj* function indicated that decreasing the *Obj* function value for a correct solution enhanced the relative value of violating intermediate due-dates, whereas overall production diminished. Table 7 compares pattern and solution statistics for correct industrial evidence and NS. The published CPU times indicated that the time for obtaining the most acceptable solution was within a time limit of 2 h.

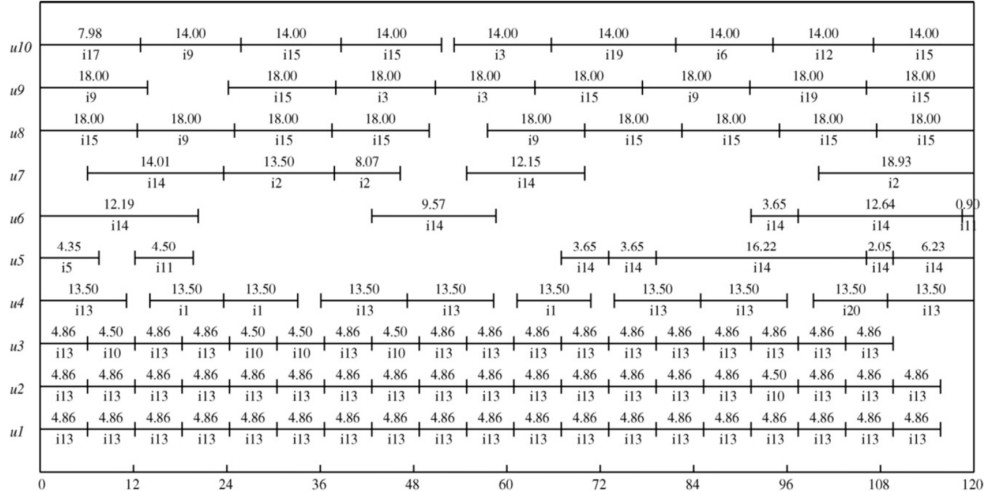

**Figure 12.** The nominal in the case-report schedule (*Obj*) = 121.37, U1–U10; units I1 to I19: missions.

**Table 7.** The model and solution statistics for the empirical report.

|  | **RS** | **NS** |
| --- | --- | --- |
| *Obj* | 105.76 | 121.37 |
| BIN variables | 930 | 930 |
| CON variables | 6161 | 6005 |
| Constraints | 22931 | 18907 |
| CPU time (s) | 5910 | 3880 |
| Nodes | 35640 | 15230 |

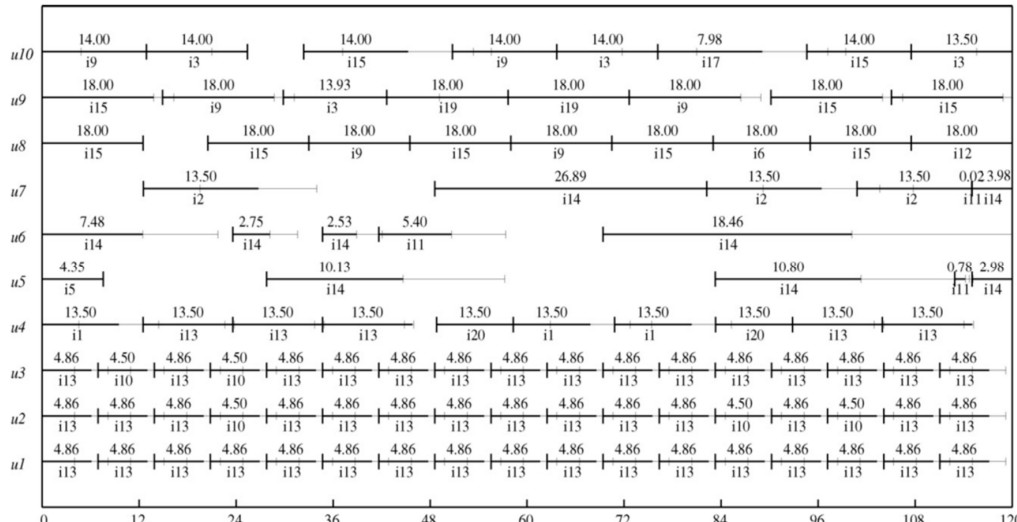

**Figure 13.** The correct time in evidence (*Obj* = 105.76, δ = 10%, 20%, U1–U10, units I1 to I19, missions. End-point of all thick horizontal lines indicated the task end time based on nominal processing duration, whereas thin vertical lines represented the end time range because of uncertainty in the processing time).

For the short-run timing problem examined in Section 5.1.1, the conservatism level observed in the resultant production timetable varied with the probability distribution function utilized for describing uncertain parameters. Comparing all types of uncertainties of each of the three uncertain constraints with various problem parameters for the example problem above revealed that general discrete distributions have consistently been the least conservative, yielding the most reasonable *Obj* function values. Additionally, binomial and Poisson distributions have consistently yielded the most conservative outputs, yielding the arrant *Obj* function values, and other kinds of distributions—that is, bounded, uniformed, and uniform distributions. Nonetheless, relative outputs for the three distributions strongly depended on the value of the level of reliability, κ. Therefore, when κ increased, indicating greater reliability, normal distribution became less conservative than the other two. For a large κ value, the normal distribution would be less conserved than the limited distribution.

Additionally, even though it is possible to use the robust optimization protocol formulation for modelling the uncertainty in diverse MILP problems, several constraints of our approach—as an example, a series of probability distribution functions—could just be utilized for limitations containing a single uncertain variable; that is, uniform, binomial, and Poisson. Such a condition would be caused by limitations in the probability theory and not the introduced formulation [83,84]. Moreover, it has not been possible for the robust optimization protocol formulation to analyze the dependent uncertain parameters connected via general non-linear expressions; however, it could be used for linear dependence on uncertain parameters. Ultimately, our formulation could control the uncertainties in the linear constraints; therefore, further studies should address such issues.

### 5.3. Exploring the Limitations of PDF for Single Uncertain Variables

The probability distribution function (PDF) of a single uncertain variable has limitations that should be taken into account. Firstly, the PDF is influenced by assumptions made about the variable and its distribution. If these assumptions are incorrect or biased, the resulting PDF may not accurately represent the true distribution. For example, assuming a normal distribution for a variable that is actually skewed could lead to misleading results.

Secondly, while the PDF provides information about the likelihood of different values, it may not fully capture the uncertainty associated with the variable. Additional measures such as confidence intervals or quantiles may be needed to obtain a more comprehensive

understanding of the uncertainty. For instance, a PDF may indicate the most likely value but fail to convey the range of possible outcomes.

Thirdly, limited information or insufficient data about the variable can result in imprecise or incomplete PDFs. If there is a scarcity of data, estimating the parameters of the distribution becomes challenging and can lead to less reliable PDFs. For example, if only a small sample size is available, the estimated PDF may not adequately represent the true population distribution.

Moreover, the assumption of independence in the PDF can be unrealistic as variables often exhibit dependencies. Ignoring these dependencies can introduce inaccuracies into the representation of uncertainty. For instance, in financial modeling, assuming independent returns for stocks may overlook correlation and result in flawed risk assessments.

Additionally, different PDFs can have similar shapes or produce the same moments (mean, variance, etc.), making it difficult to uniquely identify the true underlying distribution based solely on the PDF. This ambiguity limits the interpretability and applicability of the PDF. For example, two different distributions with the same mean and variance may have distinct characteristics that affect their practical implications.

Furthermore, while the PDF describes the past or current uncertainty of a variable, it does not necessarily possess strong predictive power for future outcomes. Extrapolating the PDF to forecast future events involves additional uncertainties and assumptions that can impact accuracy. For instance, a PDF describing historical stock prices may not reliably predict future stock market movements.

Lastly, determining the PDF of a variable can be computationally complex, particularly for complex distributions or high-dimensional problems. In such cases, approximations or simplifications may be necessary, potentially affecting the accuracy of the resulting PDF. For example, in Bayesian inference, obtaining the exact PDF may require computationally intensive methods, leading to the use of approximate techniques.

Considering these limitations is crucial when utilizing the PDF of a single uncertain variable. An awareness of the assumptions, uncertainties, and potential shortcomings involved in constructing and interpreting the PDF helps ensure its appropriate application and facilitates sound decision-making.

The present research investigates the site selection problem for an MH in Hong Kong. A robust optimization protocol algorithm is employed to deal via uncertain parameters in a mathematical model. Two scenarios of uncertainty have been investigated. Therefore, the current study proposed a novel method for addressing timing based on the problem of uncertainties using the correct optimization protocol. It is for use in applications to MILP problems and produces a "*robust*" solution that has immunity against uncertainty in coefficients and the right-side variables of disparity limitations [85–89]. This strategy could address the issue of production scheduling with market demands, unknown processing duration, and product and raw materials prices [90–92]. Additionally, the new computational results show that our innovative approach offered an effective solution to dealing with the timing issues brought on by uncertainty, producing accurate timetables and yielding useful data on the trade-offs between the conflicting aims. This technique could therefore resolve complex real-world problems with various uncertain factors as a result of its fruitful metamorphosis [10,81,93–95].

**Author Contributions:** M.H.: formal analysis, methodology, software, data curation, writing—original draft preparation, writing—review and editing, validation, funding acquisition, investigation, visualization, resources. K.K.L.: project administration, supervision. All authors have read and agreed to the published version of the manuscript.

**Funding:** This research received no external funding.

**Data Availability Statement:** Not applicable.

**Conflicts of Interest:** The authors declare no conflict of interest.

**Nomenclature**

| | |
|---|---|
| $a_{ij}$ | The fixed concept of the processing duration of mission ($i$) in the unit ($j$); |
| $\beta_{ij}$ | The varied concept of the processing times of the mission ($i$) in the unit ($j$), which expressed the time needed via the unit for processing 1 unit of the materials doing the task ($i$); |
| $\rho_{si}^p, \rho_{si}^c$ | The proportion of state ($s$) generated, which has been used by the mission ($i$), |
| $B(i, j, n)$ | The continuous amounts of the material performing the mission ($i$) in the unit ($j$) at the event point ($n$), |
| $dem_s$ | The market demand for the state ($s$) at the end of the time horizon, |
| $H$ | The time horizon, |
| $i \in$ | Missions, |
| $I$ | |
| $I_j$ | The mission that could be done in the unit ($j$), |
| $I_s$ | The mission that produced or consumed state ($s$), |
| $j \in$ | Units, |
| $J$ | |
| $J_i$ | The units appropriate to conduct the mission ($i$), |
| $n \in$ | The event points, which represented a task starting point; |
| $N$ | |
| $p_s$ | The price of reported ($s$), |
| $Profit$ | The continuous overall profit, |
| $s \in S$ | State, |
| $S_p$ | The states that corresponded to the resulting products, |
| $S_r$ | The states that corresponded to the raw materials, |
| $ST(s, n)$ | The continuous value of state ($s$) at the event point ($n$), |
| $ST_s^{\max}$ | The available Max storage for reported ($s$), |
| $STF(s)$ | The continuous final value of reported ($s$) at the end of time horizon, |
| $STI(s)$ | The continuous initial value of reported ($s$) at the starting of the time horizon, |
| $T^s(i, j, n)$ | The continuous time, at the mission ($i$) reported in the unit ($j$) at the event point ($n$), |
| $T^f(i, j, n)$ | The continuous time, at the mission ($i$) finished in the unit ($j$) whereas it started at the event point ($n$), |
| $V_{ij}^{\min}$ | The Min value of the material processed via the mission ($i$) needed for starting the operating unit ($j$), |
| $V_{ij}^{\max}$ | The Max capacity of the unit ($j$) while processing the mission ($i$); |
| $wv(i, n)$ | Binary, whether the mission ($i$) started at the event point ($n$). |

**Appendix A. Continuous-Time Procedure Scheduling Formulation**

Floudas et al. (2004, 2005) and Heydari et al. (2021) introduced the continuous time process scheduling formulation [68,69,87].

**Constraints:**

Allocation constraints

$$\sum_{i \in I_j} wv(i, n) \le 1, \qquad \forall_j \in J, \ n \in N \tag{A1}$$

$$V_{ij}^{\min} wv(i, n) \le B(i, j, n) \le V_{ij}^{\max} wv(in), \qquad \forall_i \in I, \ j \in J_i, \ n \in N \tag{A2}$$

Storage constraints

$$ST(s, n) \le ST_s^{\max}, \qquad \forall_s \in S, \ n \in N \tag{A3}$$

Material balances

$$ST(s,n) = STI(s) + \sum_{\in I_s} \rho^c_{si} \sum_{j \in J_i} B(i,j,n), \quad \forall_s \in S, \quad n \in N, n = 1$$
$$ST(s,n) = ST(s,n-1) + \sum_{i \in I_s} \rho^c_{si} \sum_{j \in J_i} B(i,j,n-1) + \sum_{i \in I_s} \rho^c_{si} \sum_{j \in J_i} B(i,j,n), \quad \forall_s \in S, \quad n \in N \tag{A4}$$
$$ST(s) = ST(s,n) + \sum_{i \in I_s} \rho^c_{si} \sum_{j \in J_i} B(i,j,n), \quad \forall_s \in S, \quad n \in N, \; n = N$$

Demand constraints

$$STF(s) \geq dem_s, \quad \forall_s \in S \tag{A5}$$

Duration constraints

$$T^f(i,j,n) = T^s(i,j,n) + a_{ij} wv(i,\,n) + \beta_{ij} B(i,j,n), \quad \forall_i \in I, \; j \in J_i, \; n \in N \tag{A6}$$

Sequence constraints: same mission in the same unit

$$T^s(i,j,n+1) \geq T^f(i,j,n), \qquad \forall_i \in I, \; j \in J_i, \; n \in N, \; n \neq N \tag{A7}$$

Sequence constraints: various mission in a similar unit

$$T^s(i,j,n+1) \geq T^f\left(i',j,n\right) - H\left[1 - wv\left(i',n\right)\right], \qquad \forall_j \in J, \; i,i' \in I_j, \; i,i' \in I_j, \; i \neq i', \; n \in N, \; n \neq N \tag{A8}$$

Sequence constraints: various mission in diverse units

$$T^s(i,j,n+1) \geq T^f\left(i',j',n\right) - H\left[1 - wv\left(i',n\right)\right], \qquad \forall_{j,j'} \in J, \; i,i' \in I_{j'}, \; i \neq i', \; n \in N, \; n \neq N \tag{A9}$$

Time horizon constraints

$$T^f(i,j,n) \leq H, \qquad \forall_i \in I, \; j \in J_i, \; n \in N$$
$$T^s(i,j,n) \leq H, \qquad \forall_i \in I, \; j \in J_i, \; n \in N \tag{A10}$$

**Objective function:**

$$Max \quad Profit = \sum_{s \in S_p} p_s.STF(s) - \sum_{s \in S_r} p_s.STI(s) \tag{A11}$$

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
