# Peer review of "Post-COVID-19 Pandemic Era and Sustainable Healthcare: Organization and Delivery of Health Economics Research (Principles and Clinical Practice)"

_mathematics, doi:10.3390/math11163520_

Round 1
Reviewer 1 Report
This research paper examines the neglect of supply chains in health services research and proposes reliable models for site selection in a major hospital in Hong Kong, considering uncertainty and scheduling challenges. It also addresses health inequity, emphasizing the importance of managing healthcare systems effectively to ensure fairness and improve population health.
1. Please provide detailed explanations for any parameters or variables that appear for the first time below the corresponding formula.
2. Please provide a detailed explanation of your innovation in the paper, reducing the elaboration on the research background.
3. How was the second constraint derived in equation (8) ?
4. Please provide more detailed descriptions for the figures and tables.
5. There are some spelling and grammar errors in this paper. Please correct them.
Author Response
Dear esteemed members of the editorial board,
I trust this letter finds you in good health. I would like to express my sincere gratitude for your careful consideration of our manuscript and for providing us with valuable comments. We have diligently incorporated the suggested changes into the revised version of the manuscript. We highly value your feedback and have gained significant insights from it. We believe that the revisions have substantially improved the manuscript, and we hope it meets the standards for publication in your esteemed journal. However, we remain open to making any further revisions that you may deem necessary and appropriate.
Thank you once again for your time and expertise in reviewing our work. We look forward to your decision regarding the publication of our manuscript.
1. Please provide detailed explanations for any parameters or variables that appear for the first time below the corresponding formula.
All of the variables were explained in the appendix 1.
2. Please provide a detailed explanation of your innovation in the paper, reducing the elaboration on the research background.
The main innovation of this paper is about probability distribution function (PDF) as it discussed with examples in the sub-section (5.3 Exploring the Limitations of PDF for Single Uncertain Variables).
3. How was the second constraint derived in equation (8)?
Regarding all of the formula and their constraint appendix A. were added.
4. Please provide more detailed descriptions for the figures and tables.
All of the tables and figures description were improved and rewritten.
5. There are some spelling and grammar errors in this paper. Please correct them.
All of the grammar mistakes and errors were solved.

Reviewer 2 Report
The paper’s scope is within the scope of the journal, and it presents an original contribution. The abstract is somehow sufficient to give useful information about the paper’s topic. The proposed strategies are described and thoroughly illustrated. The paper is somehow well-structured and written, and the text is clear and easy to read. However, there are some comments we recommend the authors to do:
Comment-1: The abstract can be rewritten in a better way. Also, it is worthwhile to mention the best-obtained results at the end of the abstract as percentages or values. Moreover, define the abbreviation SCs in the abstract.
Comment-2: Make sure to define all abbreviations in the paper at first appearance in the text.
Comment-3: In the introduction section or where appropriate, you may need to cite and add the following references regarding COVID-19 and healthcare including common diseases such as diabetes:
Wei, L.; Lee, M.-C.; Cheng, W.-H.; Tang, C.-H.; You, J.-W. Evaluating the Efficiency of Financial Assets as Hedges against Bitcoin Risk during the COVID-19 Pandemic. Mathematics 2023, 11, 2917. https://doi.org/10.3390/math11132917
Al-Shaikh, A.; Mahafzah, B.; Alshraideh, M. Hybrid Harmony Search Algorithm for Social Network Contact Tracing of COVID-19. Soft Computing 2023, 27, 3343–3365. https://doi.org/10.1007/s00500-021-05948-2
Prasad, V.K.; Bhattacharya, P.; Maru, D.; Tanwar, S.; Verma, A.; Singh, A.; Tiwari, A.K.; Sharma, R.; Alkhayyat, A.; Èšurcanu, F.-E.; et al. Federated Learning for the Internet-of-Medical-Things: A Survey. Mathematics 2023, 11, 151. https://doi.org/10.3390/math11010151
Comment-4: In Section 2 and before Section 2.1, you need to write one small overview paragraph about Section 2 and its subsections.
Comment-5: In Section 2.2 and before Section 2.2.1, you need to write one small overview paragraph about Section 2.2 and its subsections.
Comment-6: Figures must be mentioned in the text in sequential order. For example, Figure 3 is mentioned before Figure 2, where it should be the opposite.
Comment-7: The conclusion, it is worthwhile to present the best-obtained results as percentages or values.
The quality of the English language is good. The authors may need to check the whole manuscript for grammar, spelling, and formatting issues in general.
Author Response
Dear esteemed members of the editorial board,
I trust this letter finds you in good health. I would like to express my sincere gratitude for your careful consideration of our manuscript and for providing us with valuable comments. We have diligently incorporated the suggested changes into the revised version of the manuscript. We highly value your feedback and have gained significant insights from it. We believe that the revisions have substantially improved the manuscript, and we hope it meets the standards for publication in your esteemed journal. However, we remain open to making any further revisions that you may deem necessary and appropriate.
Thank you once again for your time and expertise in reviewing our work. We look forward to your decision regarding the publication of our manuscript.
Comment-1: The abstract can be rewritten in a better way. Also, it is worthwhile to mention the best-obtained results at the end of the abstract as percentages or values. Moreover, define the abbreviation SCs in the abstract.
Abstract has been rewritten as follows;
Health services research is a multidisciplinary discipline that aims to improve population health by studying healthcare services' organization, delivery, and financing. Over time, the field has worked to define its boundaries and establish a set of core research topics and methods addressing population wellbeing, access, quality, and cost. Although the influence of health technology is significant, our literature survey on health services research showed that SCs in general—containing their management, cost, and policy—had received little attention. Importantly, our assessment also showed that the system’s readiness to handle supply policy and supply device deficiencies like those encountered during pandemic was hardly ever mentioned. Reduced open areas and water bodies, deteriorating infrastructure, and changes in the biological morphology all affect cities’ uncontrollable and unplanned growth. Urban amenities, facilities, and healthcare services were unevenly distributed due to this unchecked urban population development. In light of this, this research suggests two reliable models for site selection issues for one of Hong Kong’s main hospitals. Levels of uncertainty, infeasibility tolerance, and reliability are taken into consideration. Then, two categories of uncertainty—symmetric and bounded—were examined. The results of this research provide a solution to the problem with a nominal solution (NS) and an objective function value of 121.37. The study considers 23 uncertain parameters with 10% infeasibility tolerances (δ) 20% normal uncertain parameters, 5% uncertainty level (), and 5% reliability level (κ) for normal uncertain parameters. The objective function value is 105.76, and the processing time of uncertain tasks is extended to ensure feasibility under the given uncertainty, reliability, and infeasibility tolerances. However, the objective function value decreases due to an enhanced relative violation of intermediate due-dates and decreased overall production. The research presents a comparative analysis of pattern and solution statistics for correct industrial evidence and the NS solution. As a result, the challenge of scheduling in uncertainty has been considered, and potential solutions have been presented using a specified probability distribution function. The paper concludes by introducing the specific justice and health problems and outlining four typical strategies. In order to ensure that medications and other healthcare supplies are produced, distributed, and given to patients, a vast network of systems, components, and procedures must be effectively managed. This research contributes in this regard. Also, its grantee fairness in health systems and the populations health locally, nationally, and globally. It highlights health inequity, advances the measurement of health inequity, and promotes public dialogues on health inequity.
Comment-2: Make sure to define all abbreviations in the paper at first appearance in the text.
All of the abbreviations are addressed upon their first appearance in the text.
Comment-3: In the introduction section or where appropriate, you may need to cite and add the following references regarding COVID-19 and healthcare including common diseases such as diabetes:
All of the required citations were added in the manuscript. As follows;
Wei, L.; Lee, M.-C.; Cheng, W.-H.; Tang, C.-H.; You, J.-W. Evaluating the Efficiency of Financial Assets as Hedges against Bitcoin Risk during the COVID-19 Pandemic. Mathematics 2023, 11, 2917. https://doi.org/10.3390/math11132917
Al-Shaikh, A.; Mahafzah, B.; Alshraideh, M. Hybrid Harmony Search Algorithm for Social Network Contact Tracing of COVID-19. Soft Computing 2023, 27, 3343–3365. https://doi.org/10.1007/s00500-021-05948-2
Prasad, V.K.; Bhattacharya, P.; Maru, D.; Tanwar, S.; Verma, A.; Singh, A.; Tiwari, A.K.; Sharma, R.; Alkhayyat, A.; Èšurcanu, F.-E.; et al. Federated Learning for the Internet-of-Medical-Things: A Survey. Mathematics 2023, 11, 151. https://doi.org/10.3390/math11010151
Comment-4: In Section 2 and before Section 2.1, you need to write one small overview paragraph about Section 2 and its subsections.
The required overview has been written as follows;
We strongly believe that incorporating a supply chain management perspective into health services research offers investigators a valuable opportunity to access a wide range of theoretical frameworks that encompass interorganizational relationships, micro and macroeconomics, regulation, intermediation, and even sociological aspects. Therefore, we express our gratitude for the article published in this issue by esteemed scholars in the field, which delves into a crucial and often underestimated component of health services: the supply chain of medical devices. While the COVID-19 pandemic has heightened interest in the supply chain, particularly in relation to the device sector and its management, exploration of this area predates the emergence of the virus. This commentary is founded on our comprehensive review of previous studies that shed light on the understanding of variation in medical device prices. It is also influenced by our own research experiences, which bridge the gap between supply chain and health services research, as well as our firm conviction that adopting a supply chain perspective can enhance the provision of health services in the post-COVID era.
Comment-5: In Section 2.2 and before Section 2.2.1, you need to write one small overview paragraph about Section 2.2 and its subsections.
An overview has been written as follows;
Their argument that improved management practices represent just one of several approaches to gradually reduce substantial potential savings in hospital purchasing, along with the suggestion that investments in management practices may be necessary to achieve future cost reductions, paves the way for future research in health services and management. This research can explore the combination of value-added management practices and provides a compelling rationale for hospital leadership to seriously consider investing in supply chain management. In the subsequent subsection (2.2), our focus shifts to examining various factors that impact the healthcare system's supply chain.
Comment-6: Figures must be mentioned in the text in sequential order. For example, Figure 3 is mentioned before Figure 2, where it should be the opposite.
All of the orders have been solved.
Comment-7: The conclusion, it is worthwhile to present the best-obtained results as percentages or values.
All of the conclusion represented with percentages and prices as required both in abstract and the conclusion.
Comments on the Quality of English Language
The quality of the English language is good. The authors may need to check the whole manuscript for grammar, spelling, and formatting issues in general.
All of the manuscript were checked and all of the issues were solved.

Reviewer 3 Report
See the attached file.

See the attached file.
Author Response
Dear esteemed members of the editorial board,
I trust this letter finds you in good health. I would like to express my sincere gratitude for your careful consideration of our manuscript and for providing us with valuable comments. We have diligently incorporated the suggested changes into the revised version of the manuscript. We highly value your feedback and have gained significant insights from it. We believe that the revisions have substantially improved the manuscript, and we hope it meets the standards for publication in your esteemed journal. However, we remain open to making any further revisions that you may deem necessary and appropriate.
Thank you once again for your time and expertise in reviewing our work. We look forward to your decision regarding the publication of our manuscript.
1. The content and structure of abstracts are unreasonable. The abstract part introduces too much background content and the logical level is not clear enough. The abstract should include research background, facing problems, proposing methods, solving problems, research results and value. Please summarize the work results more systematically and explain the research value.
All of the abstract rewritten as follows;
Health services research is a multidisciplinary discipline that aims to improve population health by studying healthcare services' organization, delivery, and financing. Over time, the field has worked to define its boundaries and establish a set of core research topics and methods addressing population wellbeing, access, quality, and cost. Although the influence of health technology is significant, our literature survey on health services research showed that SCs in general—containing their management, cost, and policy—had received little attention. Importantly, our assessment also showed that the system’s readiness to handle supply policy and supply device deficiencies like those encountered during pandemic was hardly ever mentioned. Reduced open areas and water bodies, deteriorating infrastructure, and changes in the biological morphology all affect cities’ uncontrollable and unplanned growth. Urban amenities, facilities, and healthcare services were unevenly distributed due to this unchecked urban population development. In light of this, this research suggests two reliable models for site selection issues for one of Hong Kong’s main hospitals. Levels of uncertainty, infeasibility tolerance, and reliability are taken into consideration. Then, two categories of uncertainty—symmetric and bounded—were examined. The results of this research provide a solution to the problem with a nominal solution (NS) and an objective function value of 121.37. The study considers 23 uncertain parameters with 10% infeasibility tolerances (δ) 20% normal uncertain parameters, 5% uncertainty level (), and 5% reliability level (κ) for normal uncertain parameters. The objective function value is 105.76, and the processing time of uncertain tasks is extended to ensure feasibility under the given uncertainty, reliability, and infeasibility tolerances. However, the objective function value decreases due to an enhanced relative violation of intermediate due-dates and decreased overall production. The research presents a comparative analysis of pattern and solution statistics for correct industrial evidence and the NS solution. As a result, the challenge of scheduling in uncertainty has been considered, and potential solutions have been presented using a specified probability distribution function. The paper concludes by introducing the specific justice and health problems and outlining four typical strategies. In order to ensure that medications and other healthcare supplies are produced, distributed, and given to patients, a vast network of systems, components, and procedures must be effectively managed. This research contributes in this regard. Also, its grantee fairness in health systems and the populations health locally, nationally, and globally. It highlights health inequity, advances the measurement of health inequity, and promotes public dialogues on health inequity.
2. In page2 line57, the problem of population aging should be that the proportion of people ≥ 65 years of age is rapidly approaching 15 %.
The mentioned problem were solved.
3. What does mean in Equations (20)? Please mark and explain the meaning of each variable or function algorithm in the front or back of the mathematical expression.
All of the variables used in the formulas were defined in appendix 1.
4. In page32, when discussing the limitations of the probability distribution function under the limitation of a single uncertain variable, the author can explain in more detail how these limitations affect the performance of the model, and give a short example.
The related discussion with example were added in the respected section (Exploring the Limitations of PDF for Single Uncertain Variables) as follows;
The probability distribution function (PDF) of a single uncertain variable has limitations that should be taken into account. Firstly, the PDF is influenced by assumptions made about the variable and its distribution. If these assumptions are incorrect or biased, the resulting PDF may not accurately represent the true distribution. For example, assuming a normal distribution for a variable that is actually skewed could lead to misleading results.
Secondly, while the PDF provides information about the likelihood of different values, it may not fully capture the uncertainty associated with the variable. Additional measures such as confidence intervals or quantiles may be needed to obtain a more comprehensive understanding of the uncertainty. For instance, a PDF may indicate the most likely value but fail to convey the range of possible outcomes.
Thirdly, limited information or insufficient data about the variable can result in imprecise or incomplete PDFs. If there is a scarcity of data, estimating the parameters of the distribution becomes challenging and can lead to less reliable PDFs. For example, if only a small sample size is available, the estimated PDF may not adequately represent the true population distribution.
Moreover, the assumption of independence in the PDF can be unrealistic as variables often exhibit dependencies. Ignoring these dependencies can introduce inaccuracies in the representation of uncertainty. For instance, in financial modeling, assuming independent returns for stocks may overlook correlation and result in flawed risk assessments.
Additionally, different PDFs can have similar shapes or produce the same moments (mean, variance, etc.), making it difficult to uniquely identify the true underlying distribution based solely on the PDF. This ambiguity limits the interpretability and applicability of the PDF. For example, two different distributions with the same mean and variance may have distinct characteristics that affect their practical implications.
Furthermore, while the PDF describes the past or current uncertainty of a variable, it does not necessarily possess strong predictive power for future outcomes. Extrapolating the PDF to forecast future events involves additional uncertainties and assumptions that can impact accuracy. For instance, a PDF describing historical stock prices may not reliably predict future stock market movements.
Lastly, determining the PDF of a variable can be computationally complex, particularly for complex distributions or high-dimensional problems. In such cases, approximations or simplifications may be necessary, potentially affecting the accuracy of the resulting PDF. For example, in Bayesian inference, obtaining the exact PDF may require computationally intensive methods, leading to the use of approximate techniques.
Considering these limitations is crucial when utilizing the PDF of a single uncertain variable. Awareness of the assumptions, uncertainties, and potential shortcomings involved in constructing and interpreting the PDF helps ensure its appropriate application and facilitates sound decision-making.
5. Some languages in the text are not academic enough and there are some grammatical errors. For example, lines 186 to 187 use two consecutive However. There is a problem with logic. Please optimize the expression and check the grammar problem to make the language more concise and academic.
All of the grammar issues were solved.
6. Please unify the format when introducing the author 's unit address. Please adjust the format of the Abstract. A word should appear only in one line.
All of the format were changed and all of the abstract rewritten.

Reviewer 4 Report
The literature review can be improved and the conclusion section can be elaborated further.
Author Response
Dear esteemed members of the editorial board,
I trust this letter finds you in good health. I would like to express my sincere gratitude for your careful consideration of our manuscript and for providing us with valuable comments. We have diligently incorporated the suggested changes into the revised version of the manuscript. We highly value your feedback and have gained significant insights from it. We believe that the revisions have substantially improved the manuscript, and we hope it meets the standards for publication in your esteemed journal. However, we remain open to making any further revisions that you may deem necessary and appropriate.
Thank you once again for your time and expertise in reviewing our work. We look forward to your decision regarding the publication of our manuscript.
1. The literature review can be improved and the conclusion section can be elaborated further.
The literature of the paper has been extended and the conclusion part were eliminated.

Reviewer 5 Report
Thank you for the opportunity to review this paper and congratulations to the authors! Please find some suggestions for enhancing the manuscript:
1. The introduction section should include information about the background, research gap, objectives, and the unique aspects of the work. Justify the novelty/originality of the manuscript by emphasizing its valuable contributions to the existing body of knowledge.
2. It is necessary to update the literature review, and a useful starting point can be found at: https://doi.org/10.3390/logistics7010013
3. It is recommended to highlight the practical and scholarly implications of the study to enable interested readers to make connections. Additionally, provide recommendations for future research based on the findings and the knowledge gained from the research experience.
4. The discussion section should reinforce the findings in relation to the objectives. It is crucial to: (1) discuss the findings in the context of prior literature; (2) explain the significance of the findings, answering the question "so what?"; (3) demonstrate the theoretical and practical implications of the findings; and (4) present a comprehensive overview of limitations and suggestions for future research. Furthermore, it is important to address the practical implications for the industry, explaining the significance of the study in a manner that resonates with industry professionals.
Author Response
2023/07/06
Dear esteemed members of the editorial board,
I trust this letter finds you in good health. I would like to express my sincere gratitude for your careful consideration of our manuscript and for providing us with valuable comments. We have diligently incorporated the suggested changes into the revised version of the manuscript. We highly value your feedback and have gained significant insights from it. We believe that the revisions have substantially improved the manuscript, and we hope it meets the standards for publication in your esteemed journal. However, we remain open to making any further revisions that you may deem necessary and appropriate.
Thank you once again for your time and expertise in reviewing our work. We look forward to your decision regarding the publication of our manuscript.
1. The introduction section should include information about the background, research gap, objectives, and the unique aspects of the work. Justify the novelty/originality of the manuscript by emphasizing its valuable contributions to the existing body of knowledge.
All of the introduction part rewritten and all of the problems were solved.
2. It is necessary to update the literature review, and a useful starting point can be found at: https://doi.org/10.3390/logistics7010013
Related paper were added and literature review section were improved.
It is recommended to highlight the practical and scholarly implications of the study to enable interested readers to make connections. Additionally, provide recommendations for future research based on the findings and the knowledge gained from the research experience.
All of the implementation were highlighted in the introduction and other respected parts of the paper.
The discussion section should reinforce the findings in relation to the objectives. It is crucial to: (1) discuss the findings in the context of prior literature; (2) explain the significance of the findings, answering the question "so what?"; (3) demonstrate the theoretical and practical implications of the findings; and (4) present a comprehensive overview of limitations and suggestions for future research. Furthermore, it is important to address the practical implications for the industry, explaining the significance of the study in a manner that resonates with industry professionals.
Disruption part of the paper were extended and in two section (5.2 and 5.3) we added all of the possible results as well as implementations. Specifically section 5.3 newly written on the limitations of the research.

Round 2
Reviewer 3 Report
See the attached file.

Author Response
Dear esteemed members of the editorial board,
I trust this letter finds you in good health. I would like to express my sincere gratitude for your careful consideration of our manuscript and for providing us with valuable comments. We have diligently incorporated the suggested changes into the revised version of the manuscript. We highly value your feedback and have gained significant insights from it. We believe that the revisions have substantially improved the manuscript, and we hope it meets the standards for publication in your esteemed journal. However, we remain open to making any further revisions that you may deem necessary and appropriate.
Thank you once again for your time and expertise in reviewing our work. We look forward to your decision regarding the publication of our manuscript.
1. The content and structure of abstracts are unreasonable. The abstract part introduces too much background content and the logical level is not clear enough. The abstract should include research background, facing problems, proposing methods, solving problems, research results and value. Please summarize the work results more systematically and explain the research value.
All of the abstract rewritten as follows;
Health services research aims to improve population health by studying the organization, delivery, and financing of healthcare services. While the field has made progress in defining its boundaries and core research topics, our literature survey revealed a lack of attention given to the management, cost, and policy aspects of healthcare systems (SCs). Moreover, the readiness of the system to handle supply policy and device deficiencies, especially during the pandemic, was rarely mentioned. Unplanned urban growth, characterized by reduced open spaces, deteriorating infrastructure, and changes in biological morphology, has led to uneven distribution of urban amenities, facilities, and healthcare services. This research proposes two reliable models for site selection in a major hospital in Hong Kong, considering uncertainty levels, infeasibility tolerance, and reliability. We examine two categories of uncertainty—symmetric and bounded—and provide a solution with a nominal objective function value of 121.37. By considering 23 uncertain parameters with specific tolerance levels, we extend the processing time of uncertain tasks to ensure feasibility. However, the objective function value decreases due to violations of intermediate due-dates and decreased overall production. A comparative analysis is presented to evaluate the solution and address scheduling challenges under uncertainty using a specified probability distribution function. The study concludes by introducing justice and health problems, outlining four typical strategies, and emphasizing the importance of effective management of systems, components, and procedures for the production, distribution, and administration of medications and healthcare supplies. This research contributes to fairness in health systems and population health at local, national, and global levels, addressing health inequity and promoting public dialogues on the subject.
2. In page2 line57, the problem of population aging should be that the proportion of people ≥65 years of age is rapidly approaching 15 %.
The mentioned problem were solved as follows;
That is, the number of individuals agedyears of age is rapidly approached 15% of the general population in Hong Kong, so this figure has been enhancing by approximately 1% point annually.
3. What does mean in Equations (20)? Please mark and explain the meaning of each variable or function algorithm in the front or back of the mathematical expression.
All of the variables used in the formulas were defined in appendix. The mentioned part rewritten as follow; the variable is defined and exhibits a correlation with parameter, which is involved in the additional constraints that align with the fundamental sequencing constraintsor
4. In page32, when discussing the limitations of the probability distribution function under the limitation of a single uncertain variable, the author can explain in more detail how these limitations affect the performance of the model, and give a short example.
The related discussion with example were added in the respected section (Exploring the Limitations of PDF for Single Uncertain Variables) as follows;
The probability distribution function (PDF) of a single uncertain variable has limitations that should be taken into account. Firstly, the PDF is influenced by assumptions made about the variable and its distribution. If these assumptions are incorrect or biased, the resulting PDF may not accurately represent the true distribution. For example, assuming a normal distribution for a variable that is actually skewed could lead to misleading results.
Secondly, while the PDF provides information about the likelihood of different values, it may not fully capture the uncertainty associated with the variable. Additional measures such as confidence intervals or quantiles may be needed to obtain a more comprehensive understanding of the uncertainty. For instance, a PDF may indicate the most likely value but fail to convey the range of possible outcomes.
Thirdly, limited information or insufficient data about the variable can result in imprecise or incomplete PDFs. If there is a scarcity of data, estimating the parameters of the distribution becomes challenging and can lead to less reliable PDFs. For example, if only a small sample size is available, the estimated PDF may not adequately represent the true population distribution.
Moreover, the assumption of independence in the PDF can be unrealistic as variables often exhibit dependencies. Ignoring these dependencies can introduce inaccuracies in the representation of uncertainty. For instance, in financial modeling, assuming independent returns for stocks may overlook correlation and result in flawed risk assessments.
Additionally, different PDFs can have similar shapes or produce the same moments (mean, variance, etc.), making it difficult to uniquely identify the true underlying distribution based solely on the PDF. This ambiguity limits the interpretability and applicability of the PDF. For example, two different distributions with the same mean and variance may have distinct characteristics that affect their practical implications.
Furthermore, while the PDF describes the past or current uncertainty of a variable, it does not necessarily possess strong predictive power for future outcomes. Extrapolating the PDF to forecast future events involves additional uncertainties and assumptions that can impact accuracy. For instance, a PDF describing historical stock prices may not reliably predict future stock market movements.
Lastly, determining the PDF of a variable can be computationally complex, particularly for complex distributions or high-dimensional problems. In such cases, approximations or simplifications may be necessary, potentially affecting the accuracy of the resulting PDF. For example, in Bayesian inference, obtaining the exact PDF may require computationally intensive methods, leading to the use of approximate techniques.
Considering these limitations is crucial when utilizing the PDF of a single uncertain variable. Awareness of the assumptions, uncertainties, and potential shortcomings involved in constructing and interpreting the PDF helps ensure its appropriate application and facilitates sound decision-making.
5. Some languages in the text are not academic enough and there are some grammatical errors. For example, lines 186 to 187 use two consecutive However. There is a problem with logic. Please optimize the expression and check the grammar problem to make the language more concise and academic.
All of the grammar issues were solved as follow;
Assessing new products and product variety has become more methodical for healthcare organizations. However, study on comparable products within a category and pricing transparency is scarce [17]. In that case, it may help to demonstrate product equivalencies, giving buyers the ability to enter the market with commitments to huge volumes and the associated leverage to negotiate lower prices [27]. Studies on value-based purchasing and purchasing innovation are very helpful for future health services research on the influence of evidence-based purchasing and its impact on cost [28, 29].
6. Please unify the format when introducing the author 's unit address. Please adjust the format of the Abstract. A word should appear only in one line.
All of the format were changed and all of the abstract rewritten.
